# Expertise Can Be Helpful for Reinforcement Learning-based Macro Placement

**Chengrui Gao**[1,2]*, **Yunqi Shi**[1,2]*, **Ke Xue**[1,2]†, **Ruo-Tong Chen**[1,2], **Siyuan Xu**[3],
**Mingxuan Yuan**[3], **Chao Qian**[1,2]†, **Zhi-Hua Zhou**[1,2]
[1]State Key Laboratory of Novel Software Technology, Nanjing University, China
[2]School of Artificial Intelligence, Nanjing University, China
[3]Huawei Noah's Ark Lab, China

## Abstract

Chip placement determines the locations of electronic components on a chip layout, which directly impacts performance, power, and area (PPA) metrics, and thus is a critical step in electronic design automation (EDA). As modern chips scale to accommodate millions of components, manual placement by human experts becomes infeasible, necessitating the use of automated algorithms. Recently, reinforcement learning (RL) has emerged as a promising approach for automating macro placement, owing to its high optimization efficiency and potential for generalization. Despite their promise, existing RL-based methods often neglect the value of expert knowledge accumulated through years of engineering practice. They tend to optimize oversimplified proxy objectives, resulting in suboptimal placements that deviate significantly from expert-designed solutions. To bridge this gap, we propose a novel RL-based placement framework that integrates EDA domain expertise from two complementary perspectives: (1) *Expert Knowledge Injection*: Incorporating well-established placement knowledge, such as dataflow guidance, periphery bias, macro grouping, and I/O keepout constraints, to guide the learning process toward human-level solutions. (2) *Expert Workflow Imitation*: Emulating the post-refinement process of human experts (i.e., updating the design iteratively based on backend PPA feedback) to progressively optimize timing metrics by employing preference optimization. Experiments on the ICCAD 2015 and OpenROAD benchmarks demonstrate that our method achieves substantial improvements in PPA metrics (e.g., 32.53% in total negative slack and 7.74% in worst negative slack compared to the runner-up method on average), outperforming advanced analytical, black-box optimization, and RL-based methods.

## 1 Introduction

Chip placement determines the physical positions of macros (e.g., memory blocks with large sizes) and standard cells (e.g., logic gates with small sizes) on the device layer of integrated circuits (ICs). It plays a vital role in the electronic design automation (EDA) process, because the placement results influence the quality of the following stages of clock tree synthesis and routing, thereby significantly affecting the final power, performance, and area (PPA) metrics of chips (Kahng et al., 2011; Shi et al., 2025b;c). In classical chip placement flows, macros are placed at first manually by experienced engineers because of their significant impact on overall PPA and larger footprint (Wang et al., 2024; Xue et al., 2025). However, modern chips (such as AI accelerators) exhibit rapidly growing on-chip memory demand (Kahng & Wang, 2025; Shi et al., 2025a), leading to large numbers of memory macros and elevating the importance of automated macro placement algorithms (Patwardhan, 2020).

Reinforcement learning (RL) has emerged as an encouraging direction for automating macro placement since AlphaChip's publication on *Nature* (Mirhoseini et al., 2021; Goldie et al., 2024). It formulates chip placement as a Markov decision process (MDP): At each step, the agent places a single macro, and the environment transitions to the next state of the partially placed layout. After

---

*Equal contribution.
†Corresponding authors: {xuek,qianc}@lamda.nju.edu.cn

Table 1: Comparison of macro placement methods with respect to the incorporation of expert placement knowledge and the utilization of timing metrics.

| Methods | Type | Integrated Expert Knowledge | | | | Utilization of Timing Metrics |
|---|---|---|---|---|---|---|
| | | Dataflow | Macro Grouping | Periphery Biasing | I/O Keepout | |
| DREAMPlace (Lin et al., 2020) | Analytical | | | | | |
| DREAMPlace 4.0 (Liao et al., 2023) | Analytical + BBO | | | | | |
| AutoDMP (Agnesina et al., 2023) | Analytical + BBO | | | | | |
| IncreMacro (Pu et al., 2024) | Analytical + MILP | | | ✓ | | |
| Hier-RTLMP (Kahng et al., 2023) | BBO | ✓ | ✓ | ✓ | ✓ | |
| WireMask-BBO (Shi et al., 2023) | BBO | | | | | |
| LaMPlace (Geng et al., 2025) | BBO + Regression | | | | | ✓ |
| AlphaChip (Mirhoseini et al., 2021) | RL | | | | | |
| DeepPlace (Cheng & Yan, 2021) | RL | | | | | |
| MaskPlace (Lai et al., 2022) | RL | | | | | |
| ChiPFormer (Lai et al., 2023) | RL | | | | | |
| MaskRegulate (Xue et al., 2024) | RL | | | ✓ | | |
| EfficientPlace (Geng et al., 2024) | RL | | | | | |
| EffectivePlace (Lin et al., 2025) | RL | | ✓ | | | |
| **EXPlace (Ours)** | RL | ✓ | ✓ | ✓ | ✓ | ✓ |

that, many works have improved RL for chip placement from various perspectives, such as better state representation (Cheng & Yan, 2021; Cheng et al., 2022) and dense rewards (Lai et al., 2022; Xue et al., 2024; Ren et al., 2026), with recent works showing some better performance metrics than analytical placers (Lai et al., 2023; Geng et al., 2024; Chen et al., 2026b; Lu et al., 2026).

Despite their progress, existing RL-based methods from academia face significant barriers to deployment in real production flows. A key reason is their focus on oversimplified objectives such as macro half-perimeter wirelength (HPWL) and rectangular uniform wire density, which fail to incorporate the critical engineering know-hows (e.g., grouping macros according to the design hierarchy, and optimizing macro-level dataflow) and the complex design rules (e.g., placing macros close to the chip periphery, and keeping I/O port regions clear) that have been proven over years in the EDA industry, as shown in Table 1. These rules are crucial for downstream processes like clock tree synthesis and routing, and have become the standard expert knowledge encoded in commercial tools. This oversight makes academic RL solutions inadequate for real-world challenges, further widening the gap between academic research and industrial application.

To overcome the existing drawbacks, we propose an expert-like RL placement framework EXPlace that integrates domain **EX**pertise to help chip **Place**ment from two complementary perspectives. (1) *Expert Knowledge Injection*: EXPlace incorporates four well-established placement expertise to guide the learning process toward human-level solutions, including (a) Dataflow: Guides macros with stronger dataflow connectivity to be placed closer, improving timing path efficiency; (b) Macro Grouping: Encourages co-placement of structurally similar and highly connected macros to reduce wirelength and enhance layout regularity; (c) Periphery Biasing: Promotes macro placement near the chip periphery to avoid congestion in core routing regions and improve routability; (d) I/O Keepout: Reserves keepout regions around I/O ports to ensure buffer space and prevent routing conflicts. They are converted into state representations and dense reward designs that are readily integrated into existing RL frameworks. (2) *Expert Workflow Imitation*: Human experts typically update the design iteratively based on later-stage feedback such as timing metrics. To emulate this process, EXPlace can progressively optimize timing by preference optimization (Rafailov et al., 2023).

To validate the effectiveness of our proposed framework, we conduct comprehensive experiments on two widely used macro placement benchmarks: the ICCAD 2015 Contest (Kim et al., 2015) and the OpenROAD-flow-scripts Benchmark (Ajayi et al., 2019) Suites. We compare EXPlace against several state-of-the-art placement methods of three representative categories, i.e., analytical placers: DREAMPlace (Lin et al., 2020; Liao et al., 2023), black-box optimization (BBO) placers: Hier-RTLMP (Kahng et al., 2023) and LaMPlace (Geng et al., 2025), and RL-based placers: MaskPlace (Lai et al., 2022) and MaskRegulate (Xue et al., 2024). Experimental results demonstrate that EXPlace achieves the best average rank on all the critical metrics, including post-route wirelength (rWL), worst negative slack (WNS), and total negative slack (TNS), while maintaining minimal design rule checking (DRC) violations, directly confirming its practicality for real production flows. For example, compared to runner-up methods on the ICCAD 2015 benchmark, EXPlace achieves significant average improvements of 32.53% on TNS and 7.74% on WNS. Ablation studies also validate the effectiveness of our proposed modules.

Our contribution lies in the ability to make RL-based placement practical for industry. By comprehensive expert knowledge injection and expert workflow imitation, EXPlace not only improves PPA quality but also reduces the gap between automated layouts and human-designed ones, eliminating the need for extensive manual refinement. It also exemplifies a promising paradigm of AI for EDA: *Integrating domain expertise* can be helpful for data-driven learning methods. Note that integrating domain knowledge in the process of utilizing data, e.g., abductive learning (Zhou & Huang, 2021), has been a promising direction in AI. We have open-sourced our code at `lamda-bbo/EXPlace`, to support replication and further comparison.

## 2 PRELIMINARIES

Chip placement is a vital stage in modern chip design. Given a set of modules (macros and standard cells) to place, a netlist specifying inter-module connectivity, and a fixed-outline canvas, the chip placement algorithm is expected to determine the optimal physical locations of all movable modules on the canvas such that key design objectives (e.g., wirelength, timing, or power) governing overall chip performance are effectively optimized. In well-established design flows, this process is typically divided into two successive stages: macro placement and standard cell placement (Ajayi et al., 2019; Mirhoseini et al., 2021). Existing chip placement methods generally fall into three categories: (1) Analytical placers, which formulate placement as a continuous optimization problem with smoothed surrogate objectives and solve it via gradient-based techniques, (2) BBO placers, which search the placement solution space iteratively guided by black-box evaluation feedbacks, and (3) RL placers, which cast the chip placement process as an MDP and train an agent to sequentially place modules (See App. A.1 for a detailed discussion). Despite their ability to optimize certain proxy metrics (e.g., macro HPWL) effectively, existing methods often yield limited performance in final PPA metrics such as TNS and WNS, resulting in designs that still diverge significantly from those achieved by human experts.

Recently, there are several efforts that attempt to incorporate expert knowledge to fill this gap. For example[1], the analytical placer IncreMacro (Pu et al., 2024) designs a smoothed and differentiable periphery cost to push macros to the chip periphery, several BBO placers incorporate dataflow into placement (Vidal-Obiols et al., 2019; Lin et al., 2021; Zhao et al., 2025) to capture the hidden connections, and the RL placer MaskRegulate (Xue et al., 2024) encourages more regular placement result. Nevertheless, existing methods for integrating expert knowledge still possess several issues. (1) *Incomplete Expertise Coverage*: Most existing learning-based methods rely on oversimplified surrogates or consider only a single type of expertise, such as peripheral placement. None of these approaches comprehensively capture the full spectrum of industrial knowledge. Meanwhile, although BBO methods can accommodate a broader range of expertise, they are inherently constrained by their black-box formulations, which impede the effective modeling and optimization of expert-defined rules. (2) *Neglecting PPA feedback*: The most direct approach to optimizing sign-off performance is to replace approximate objectives with more accurate PPA feedback, such as timing metrics. However, despite the critical role of PPA evaluations, most existing methods tend to overlook their impact and instead overfit to surrogates, largely due to the high cost of PPA evaluations. In summary, how to efficiently incorporate established domain expertise into the RL framework, and how to effectively leverage feedback from later-stage evaluations such as timing analysis, remain significant open challenges. These challenges provide a clear motivation for our proposed EXPlace, which integrates four well-established placement expertise into RL and simulates the iterative timing refinement process within the chip placement pipeline.

## 3 INTRODUCING EXPERT KNOWLEDGE INTO RL

In the MDP formulation of macro placement, an agent determines the location of the next macro conditioned on the positions of those already placed. Therefore, an appropriate way to leverage expertise is to decompose high-level guidance into step-wise signals aligned with the placed layout. These step-wise signals are then integrated into both the reward and the state features of the RL agent to enable efficient optimization. Following this principle, we present our RL design along four perspectives of expertise: dataflow, macro grouping, periphery biasing, and I/O keepout, which will be introduced in detail in the following sections.

---

[1]Detailed related works about methods that incorporate expert knowledge are provided in App. A.2 due to space limitation.

### 3.1 DATAFLOW

Dataflow is a register-transfer-level (RTL) connectivity signature that reflects both the strength and direction of information flow between functional blocks, capturing the architect's design intent and providing effective guidance for macro placement to both human experts and automated algorithms (Vidal-Obiols et al., 2019; Lin et al., 2021; Kahng et al., 2023). As shown in App. D.4, dataflow conveys substantially richer information than raw netlist connectivity alone. Intuitively, macros exhibiting strong dataflow affinity should be placed in close physical proximity so as to facilitate the placement and routing of their associated standard cells. Furthermore, because data transfers are typically processed by combinational logic or staged through flip-flops, dataflow paths align naturally with

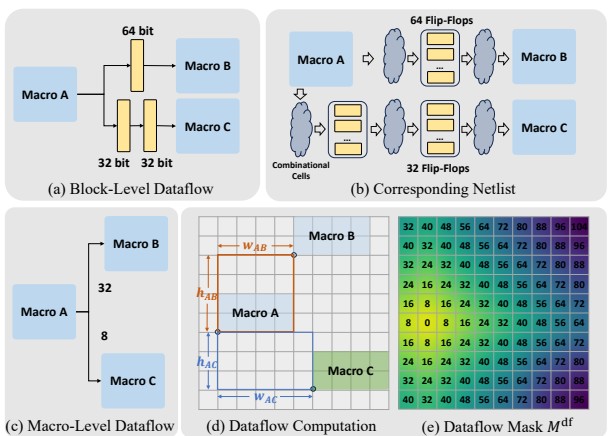

Figure 1: Illustration of dataflow and its mask. (a)-(c): Dataflow at different levels. (d)-(e): Dataflow mask of placing Macro C. Each mask value equals the dataflow weight between A and C multiplied by their Manhattan distance.

the critical timing paths evaluated for WNS and TNS, which predominantly span between macros and flip-flops (Shi et al., 2025d; Chen et al., 2026a).

We extract dataflow information at the netlist level, to which the RTL connectivity has been synthesized. For example, Fig. 1(a)-(b) depict the dataflow among macros A, B, and C. Macro A drives macro B through a 64-bit register, which is synthesized into 64 flip-flop cells together with additional combinational cells in the netlist. This structure faithfully reflects the dataflow from A to B. We then summarize the dataflow information by constructing a macro-level dataflow graph, as shown in Fig. 1(c), where combinational cells are removed, and the remaining macro-to-macro paths are aggregated into edge weights that quantify dataflow strength. Formally, the weight between macro $i$ and $j$ is computed as $w(i,j) = \sum_{p \in P_{i,j}} \frac{1}{2^{N(p)}}$, where $P_{i,j}$ is the set of paths from macro $i$ to $j$, and $N(p)$ is the number of hops (i.e., flip-flops nodes) on path $p$. That is, longer paths contribute less to the overall dataflow. To leverage dataflow guidance, we define a dataflow-weighted Manhattan distance cost,

$$\text{cost}^{\text{df}} = \sum_{i \in M} \sum_{j \in M} w(i,j) \cdot \|\text{pos}_i - \text{pos}_j\|_1, \quad (1)$$

where $M$ denotes the set of all macros, $\text{pos}_i$ and $\text{pos}_j$ denote the positions of macro $i$ and $j$, respectively. Minimizing this cost encourages macros with stronger dataflow to be placed closer together. Moreover, this cost function is composed of independent pairwise terms with the form $w(i,j) \cdot \|\text{pos}_i - \text{pos}_j\|_1$. In the sequential decision process of RL, we can sum up the pairwise costs between the new macro and those already placed to obtain the incremental cost incurred by placing the next macro, which naturally yields a dense reward signal that facilitates policy optimization. Specifically, the dataflow reward at step $t$ for placing macro $m_t$ at $(x,y)$ is

$$r_t^{\text{df}}(a_t = (x,y)|s_t) = -\sum_{i \in M_{t-1}} (w(i,m_t) + w(m_t,i)) \cdot \|\text{pos}_i - (x,y)\|_1,$$

where $M_{t-1}$ denotes the set of macros placed in the previous $t-1$ steps. Following the design principle of wire mask (Lai et al., 2022), we calculate the expected reward $r_t^{\text{df}}(x,y)$ for each grid position $(x,y)$, and obtain a two-dimensional dataflow mask $M_t^{\text{df}}$ with $M_t^{\text{df}}(x,y) = -r_t^{\text{df}}(x,y)$, as shown in Fig. 1(d)-(e). This mask quantifies the impact of each candidate position on the dataflow cost and implies a gradient-like tendency of $r_t^{\text{df}}$ across the placement canvas, which provides expressive features and is incorporated into our state encoding to facilitate dataflow-aware optimization.

## 3.2 MACRO GROUPING

As chips continue to scale and accommodate increasingly complex functionality, hierarchical design has become the prevailing paradigm for frontend engineers. Fig. 2(a) illustrates a simplified hierarchy tree. Within each hierarchy, many macros recur with similar structure; for example, when a 4 KB memory is required at hierarchy level 1, a macro compiler may select four identical 1 KB macros and concatenate them along the width dimension. These reused macros typically share an identical footprint, exhibit highly similar cell connectivity, and frequently exchange data. Consequently, experienced designers often group and co-place them. Such a grouping strategy reduces wirelength, enables regular tiling of equal-sized macros with minimal dead space, simplifies clock-tree balancing, and facilitates cleaner power delivery networks (Lin et al., 2019a; Le et al., 2023).

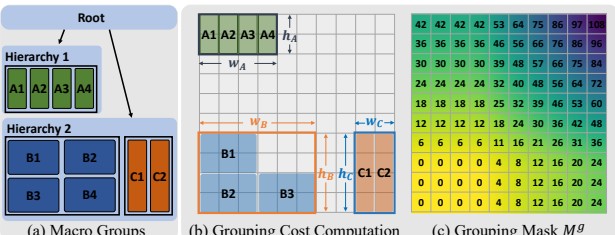

(a) Macro Groups    (b) Grouping Cost Computation    (c) Grouping Mask $M^g$

Motivated by Hier-RTLMP (Kahng et al., 2023), which groups macros by hierarchy, connection signature, and footprint, we adopt a stricter criterion that macros are grouped only if they simultaneously satisfy three conditions: Sharing over 30 nets to the same cell cluster, having the similar footprint, and belonging to the same design hierarchy. If no explicit hierarchy is available, we can infer a surrogate hierarchy from the netlist via Louvain clustering (Fogaca et al., 2020) and then apply the first two rules to form macro groups.

Figure 2: Illustration of the macro grouping mask. (a) Hierarchy tree yielding three macro groups. (b) Grouping cost defined as the area of each group's bounding box. (c) Grouping mask for placing macro B2.

To promote spatial proximity within each group, we define a grouping cost as the sum of the areas of all group bounding boxes: $\text{cost}^g = \sum_{c_i \in C} w_i \cdot h_i$, where $C$ is the set of all macro groups, $w_i$ and $h_i$ denote the width and height of the bounding box of $c_i$, respectively, as depicted in Fig. 2(b). The core idea of designing dense reward and mask for macro grouping is similar to that of dataflow. To be specific, the grouping cost also admits an incremental decomposition: Each macro is assigned to exactly one group, and when placing a macro, we only account for the incremental change in its group's bounding box. The resulting increase in bounding-box area thus serves as a natural dense reward whose cumulative sum equals the total grouping cost over the full placement. Analogous to the dataflow mask, the grouping mask $M^g$ encodes the expected reward across candidate positions, as shown in Fig. 2(c), and is incorporated into the state representation as a key feature.

## 3.3 PERIPHERY BIASING

Placing macros near the chip periphery is a well-established practice, widely adopted by both traditional placers (Kahng et al., 2023; Pu et al., 2024) and learning-based placers (Xue et al., 2024). If macros are concentrated in the core region, standard cells will be pushed outward, worsening the overall wirelength. This is because macros typically occupy the lower metal layers which are used for signal routing; consuming these layers in the core forces long detours and can induce severe routing congestion. Modern analytical global placers also require large, regular, and continuous regions for effective optimization; otherwise, they may struggle to converge. In addition, placing macros near the periphery facilitates more robust power and clock planning, which is vital for design closure.



Figure 3: Illustration of peripheral biasing mask.

To bias macros toward the periphery during placement, we define the periphery cost as $\text{cost}^{\text{peri}} = \sum_{i \in M} (\text{d}_x^i + \text{d}_y^i)$, where $M$ denotes the set of all macros, $\text{d}_x^i$ and $\text{d}_y^i$ are the distances from macro $i$ to the nearest periphery in the $x$ and $y$ directions, respectively. At each placement step, we compute a periphery mask $M^{\text{peri}}$ that evaluates the incremental periphery cost after placing the current macro at each candidate grid, as shown in Fig. 3. The sum of step-wise mask values equals the total periphery cost.

## 3.4 I/O KEEPOUT

Every design exchanges power, clocks, and signals with external environment through I/O ports located on the chip periphery. These I/O signals typically traverse long paths before reaching internal components and thereby require buffering to drive lengthy interconnects and high fanout. Meanwhile, macros are commonly placed near the periphery and occupy both area and several metal layers, thereby reducing the available resources for I/O buffering and routing. Though critical, this I/O congestion issue has not been explicitly addressed by learning-based placers to date. We mitigate this issue by reserving keepout regions around I/O ports, and defining an I/O keepout cost that penalizes macro overlap with the reserved regions: $\text{cost}^{\text{IO}} = \sum_{i \in M} \text{overlap}(i, \text{I/O regions})$, where the $\text{overlap}(\cdot)$ function measures the area of macro $i$ that intersects the keepout regions, which are pre-defined according to the I/O port positions. Since the overlap area is independent for each macro, the

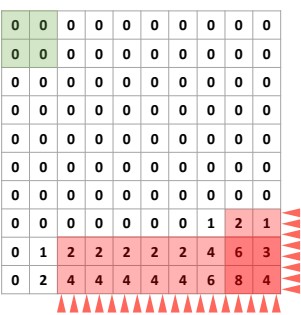

Figure 4: Illustration of I/O keepout mask.

stepwise reward and its corresponding mask can be computed directly. As shown in Fig. 4, the red triangles denote I/O ports, and the red shading indicates the I/O keepout region. For a 2×2 macro, each mask value equals the overlap area between the macro and the keepout region.

## 4 EXPERT-LIKE RL WORKFLOW

During the long-term design process of modern advanced ICs, human experts typically construct an initial, high-quality layout prototype by leveraging their own experience. This prototype is then iteratively refined through successive evaluation-and-update steps, progressively improving layout quality until the performance specifications are satisfied. Within a learning-based framework, this prototyping–refinement workflow naturally suggests a pipeline, i.e., pre-training and fine-tuning. In this section, we introduce how EXPlace instantiates this idea by (1) developing an expertise-guided RL pre-training procedure (Sec. 4.1) and (2) adopting direct preference optimization (Rafailov et al., 2023) for timing-driven fine-tuning (Sec. 4.2).

### 4.1 RL TRAINING WITH EXPERTISE GUIDANCE

Building on the pre-mentioned domain expertise, we develop EXPlace, an expertise-guided RL placer that incorporates expert masks throughout the MDP and policy design. We specifically detail the MDP for EXPlace as follows.

**State.** We define the state on a discretized placement grid $G$, where each grid site is a candidate location for macro placement. To provide the agent with sufficient context, the state comprises the following components: (1) *Canvas of current layout*, a binary map indicating placed grids and available grids. (2) *Position mask*, a binary mask marking grids that can accommodate the next macro without overlap. (3) *Expert masks*: As detailed in Sec. 3, we incorporate the dataflow mask, grouping mask, periphery mask, and I/O keepout mask, which collectively encode expert priors to guide action selection. The wiremask is also included to optimize direct macro-to-macro connections. (4) *Next-step masks*: Following MaskPlace (Lai et al., 2022), we include next-step masks to equip the agent with long-horizon awareness. This composite representation provides both instantaneous feasibility and structured expert guidance, facilitating informed, constraint-aware placement decisions.

**Action.** At time $t$, the agent selects a valid grid $a_t = (x, y)$ for placing the next macro in a pre-defined macro ordering (Lai et al., 2022) that prioritizes macros with high connectivity density and large footprints. Moreover, we design a constrained action variant based on *corner stitching*, which restricts candidate positions to corners of already placed macros or the canvas boundaries, inspired by (Ousterhout, 1984) and (Lin et al., 2019b). In this setting, the policy's selected position is snapped to the nearest corner-aligned grid. This constrained action space further enhances the periphery bias and often promotes regular patterns with reduced internal dead space. However, such strict peripheral placement is not universally desirable (Kahng et al., 2023), as it can increase wirelength and constrain placement topologies. In our experiments, we activate corner stitching for OpenROAD benchmarks, whose uniform macro shapes and clear hierarchy tend to benefit from it.

**Reward.** All the costs based on expert knowledge (introduced in Sec. 3) can be transformed into step-wise dense rewards. Specifically, each reward term is defined as the negative of its corresponding mask-derived cost and normalized using per-criterion maxima and minima. For example, the dense dataflow reward is computed by $r_t^{\mathrm{df}}(x, y) = -(M_t^{\mathrm{df}}(x, y) - M_{\min}^{\mathrm{df}})/(M_{\max}^{\mathrm{df}} - M_{\min}^{\mathrm{df}})$, where $M_{\max}^{\mathrm{df}}$ and $M_{\min}^{\mathrm{df}}$ are estimated from statistics collected in the first trajectory. The overall reward aggregates all criteria via weighted sum. In practice, we up-weight periphery biasing and grouping to encourage regular placements, and down-weight the I/O keepout constraint to allow greater flexibility (Kahng et al., 2023). The remaining weights are generally set equal.

**Policy Training.** We train the policy using proximal policy optimization (Schulman et al., 2017), where the policy network is implemented as a convolutional neural network with local and global branches to process the multi-channel state input and output an action distribution over the grid. This training leverages the dense, expert-informed rewards to efficiently learn a placement policy.

### 4.2 TIMING-DRIVEN FINE-TUNING

To bridge the gap between proxy objectives (e.g., macro HPWL) and final timing metrics (i.e., WNS and TNS), we propose a timing-driven fine-tuning procedure that directly optimizes the results of static timing analysis (STA) (Huang et al., 2020). Specifically, EXPlace identifies trajectories yielding better STA results as *preferred* samples and leverages direct preference optimization (DPO) (Rafailov et al., 2023) to shift the solution distribution accordingly. This formulation improves sample efficiency through trajectory-level evaluations and, by relying on qualitative optimization signals, helps mitigate the impact of the rugged landscape of STA results.

At each fine-tuning iteration, EXPlace samples multiple trajectories and partitions the self-generated data into preferred and rejected sets based on their STA results, thereby forming the preference pairs $D$. The policy is then fine-tuned by minimizing the following DPO loss:

$$\mathcal{L}_{\mathrm{DPO}} = -\mathbb{E}_{(\tau_w, \tau_l) \sim D}[\log \sigma(\beta \log \frac{\pi_\theta(\tau_w)}{\pi_{\mathrm{ref}}(\tau_w)} - \beta \log \frac{\pi_\theta(\tau_l)}{\pi_{\mathrm{ref}}(\tau_l)})], \tag{2}$$

where $\pi_{\mathrm{ref}}$ is the reference policy from which trajectories are sampled, $\sigma$ denotes the logistic function, $\beta$ controls the trade-off between maximizing reward and limiting deviation from $\pi_{\mathrm{ref}}$, and $\pi_\theta(\tau_w)$ denotes the likelihood of trajectory $\tau_w$, i.e., $\Pi_{(s_t, a_t) \in \tau_w} \pi_\theta(a_t | s_t)$. Intuitively, Eq. (2) increases the likelihood of preferred trajectories $\tau_w$ while decreasing that of rejected ones $\tau_l$, aligning the policy distribution with self-generated timing preferences. With repeated DPO-based self-improvement, probability mass progressively shifts toward trajectories with better timing metrics, leading to consistent improvements in TNS and WNS, which will be shown in the experiments.

## 5 EXPERIMENT

### 5.1 EXPERIMENTAL SETTINGS

**Benchmarks and Evaluation Metrics.** We evaluate our proposed method on two widely adopted benchmarks for macro placement: the ICCAD 2015 Contest Benchmark Suite (Kim et al., 2015) and the OpenROAD-flow-scripts Benchmark Suite (Ajayi et al., 2019). For the ICCAD 2015 dataset, which includes 8 placement cases (see Table 2), we employ DREAMPlace (Liao et al., 2023) for global placement and utilize EarlyGlobalRoute to evaluate post-route metrics, including routed wirelength (rWL), horizontal and vertical overflow (rOverflowH and rOverflowV), the number of violated timing paths (NVP), as well as post-route TNS and WNS. For the OpenROAD benchmark, we select 6 representative design cases (see Table 3). We first use our proposed method to place macros. Then OpenROAD is applied to perform a complete flow of global placement, clock tree synthesis, global routing, and detailed routing, to acquire the rWL, WNS, TNS, design rule checking (DRC) violations, Power, and Area, providing comprehensive and accurate assessments of sign-off PPA metrics. Note that the results of EXPlace reported in Tables 2 and 3 are obtained using the pre-trained policy without timing-driven fine-tuning, ensuring a fair comparison with other RL-based methods. The impact of fine-tuning is analyzed separately in Fig. 5.

**Compared Methods.** To thoroughly assess the effectiveness of our approach, we compare it against three representative categories of macro placement methods: (1) *Analytical placer.* We adopt

DREAMPlace 4.1.0 (Liao et al., 2023) as a baseline, which formulates placement as a gradient-optimizable problem by relaxing the HPWL objective into a differentiable surrogate. In our experiments, DREAMPlace is configured to perform mixed-size placement, simultaneously handling both macros and standard cells. (2) *BBO placers.* We evaluate two representative BBO-based methods: Hier-RTLMP (Kahng et al., 2023) and LaMPlace (Geng et al., 2025). Hier-RTLMP formulates macro placement as a bin-packing problem and applies simulated annealing guided by expert-designed heuristics to perform optimization. As it is integrated into the OpenROAD-flow-scripts, we evaluate it only on the OpenROAD benchmark. LaMPlace introduces a learned mask prediction model trained on historical placement and PPA data, which produces PPA-aware masks to guide the search process toward improved final outcomes. (3) *RL placers.* We compare our method with two advanced RL methods: MaskPlace (Lai et al., 2022) and MaskRegulate (Xue et al., 2024). They learn placement policies by mainly optimizing oversimplified proxy rewards, where MaskPlace adopts a purely data-driven paradigm optimizing HPWL, and MaskRegulate incorporates domain knowledge in the form of periphery constraints. Thus, the comparison with these two RL-based methods is particularly relevant for evaluating the benefits of our proposed expertise-driven approach. Detailed settings are provided in App. C due to space limitation.

## 5.2 MAIN RESULTS

**Results on ICCAD 2015 Contest Benchmark.** Table 2 summarizes the performance of the proposed method and baselines on the ICCAD 2015 contest benchmark. The proposed EXPlace attains the best average rank across five of the six evaluated metrics. In particular, relative to runner-up methods, EXPlace yields average improvements of 3.41% on rWL, 10.73% on NVP, 7.74% on WNS and 32.53% on TNS. Most notably, compared to LaMPlace (Geng et al., 2025), which is the overall runner-up method across all metrics, EXPlace still achieves an average improvement of 32.53% on TNS, which can substantially ease timing closure in practical design. Furthermore, compared with MaskRegulate (Xue et al., 2024), which integrates only periphery cost into RL framework, our method obtains superior overall performance, highlighting the effectiveness of comprehensively incorporating expert knowledge. The sole exception is rOverflowV, where EXPlace is slightly inferior because MaskRegulate's exclusive focus on periphery cost enforces extremely tight packing near the boundary, directly favoring this particular metric.

Table 2: Comparison of different placement methods on ICCAD 2015 contest benchmarks. The evaluation metrics include rWL (m), rOverflowH (%), rOverflowV (%), NVP ($10^3$), WNS (ns), TNS ($10^4$ ns), as well as the average ranking across all cases for overall performance assessment. The best and runner-up results are highlighted in **bold** and underline, respectively.

| Method | Metric | superblue1 | superblue3 | superblue4 | superblue5 | superblue7 | superblue10 | superblue16 | superblue18 | Avg. Rank |
|---|---|---|---|---|---|---|---|---|---|---|
| **DREAMPlace** (Liao et al., 2023) | rWL | 159 | 199 | 140 | 167 | 220 | 244 | 136 | **53.3** | 4.38 |
| | rOverflowH | 16.93% | 15.05% | 18.74% | 5.67% | 6.93% | 3.83% | 16.37% | **0.19%** | 4.50 |
| | rOverflowV | 4.72% | 7.40% | 11.60% | 2.18% | 5.24% | 2.10% | 1.93% | 0.02% | 4.62 |
| | NVP | 30.0 | 19.8 | 26.9 | 20.5 | 38.1 | 24.5 | 37.5 | 5.02 | 4.50 |
| | WNS | -111 | -108 | -106 | **-71.7** | -59.6 | -240 | -65.5 | -30.3 | 3.75 |
| | TNS | -37.3 | -25.1 | -21.8 | -17.4 | -13.7 | -41.0 | -21.0 | -1.78 | 4.12 |
| **MaskRegulate** (Xue et al., 2024) | rWL | 121 | 149 | 87.3 | 142 | 186 | 225 | 102 | 58.5 | 3.00 |
| | rOverflowH | **0.07%** | 0.37% | 0.39% | **0.11%** | 0.20% | 0.18% | **0.36%** | 0.20% | **2.00** |
| | rOverflowV | **0.02%** | 0.05% | 0.07% | 0.12% | 0.03% | 0.09% | 0.06% | **0.01%** | **1.88** |
| | NVP | **12.5** | 12.0 | 9.94 | 13.2 | 19.3 | 16.3 | 17.4 | 5.33 | 2.25 |
| | WNS | -66.4 | -135 | -63.0 | -127 | -70.4 | **-67.3** | -42.9 | -32.5 | 3.38 |
| | TNS | -12.3 | -17.4 | -11.5 | -11.6 | -18.1 | -28.1 | -14.7 | -1.95 | 2.75 |
| **MaskPlace** (Lai et al., 2022) | rWL | 139 | 175 | 105 | 172 | 214 | 212 | 115 | 63.8 | 4.12 |
| | rOverflowH | 4.61% | 8.91% | 12.00% | 2.74% | 2.41% | 0.64% | 8.82% | 0.70% | 4.12 |
| | rOverflowV | 0.39% | 0.66% | 0.52% | 0.18% | 0.56% | 0.05% | 0.16% | 0.14% | 3.88 |
| | NVP | 27.3 | 19.9 | 17.9 | 20.1 | 37.1 | 19.2 | 25.4 | 29.6 | 4.25 |
| | WNS | -77.4 | -102 | -71.9 | -169 | -93.0 | -67.8 | -50.2 | -35.3 | 4.00 |
| | TNS | -31.5 | -22.7 | -16.9 | -22.1 | -24.4 | -35.2 | -25.5 | -10.4 | 4.50 |
| **LaMPlace** (Geng et al., 2025) | rWL | 117 | **103.1** | 95.3 | 131 | 189 | 210 | 100 | 58.3 | 2.12 |
| | rOverflowH | 0.19% | **0.02%** | 1.50% | 0.99% | 0.16% | **0.06%** | 0.94% | 0.19% | 2.12 |
| | rOverflowV | 0.07% | **0.01%** | 0.10% | 0.15% | 0.04% | **0.03%** | 0.04% | 0.02% | 2.12 |
| | NVP | 14.3 | **4.24** | 13.6 | 18.1 | 18.1 | 16.7 | 18.0 | 7.48 | 2.75 |
| | WNS | -47.2 | **-76.2** | -46.6 | -117 | -59.8 | -69.9 | -37.2 | -29.6 | 2.12 |
| | TNS | -12.9 | **-3.49** | -10.7 | -10.8 | -12.0 | -33.0 | -15.3 | -3.60 | 2.50 |
| **EXPlace (Ours)** | rWL | **108** | 134 | **77.5** | 138 | **171** | **195** | **90.2** | 55.8 | **1.38** |
| | rOverflowH | 0.15% | 1.24% | **0.27%** | 0.30% | **0.01%** | 0.13% | 0.36% | 0.41% | **2.00** |
| | rOverflowV | 0.02% | 0.10% | **0.05%** | **0.02%** | **0.01%** | 0.12% | 0.07% | 0.02% | **2.00** |
| | NVP | 13.2 | 9.86 | **8.77** | **12.4** | **13.9** | **15.3** | **16.5** | **4.67** | **1.25** |
| | WNS | -58.6 | -82.3 | **-34.1** | -111 | **-41.8** | -71.6 | **-21.4** | **-25.3** | **1.75** |
| | TNS | **-8.63** | -7.80 | **-6.65** | **-8.96** | **-9.13** | **-20.9** | **-5.29** | **-1.32** | **1.12** |

**Results on OpenROAD Benchmark.** Table 3 reports the performance on the OpenROAD benchmark. Our proposed EXPlace achieves the best average ranks across all metrics, including rWL

Table 3: Comparison experiments on OpenROAD benchmarks. We report rWL (m), TNS (ns), WNS (ns), DRC, Power (W), Cell Area ($\times 10^4 \mu m^2$) and average ranking.

| Method | Metric | ariane133 | ariane136 | bp | bp_be | bp_fe | swerv_wrapper | Avg. Rank |
|---|---|---|---|---|---|---|---|---|
| **DREAMPlace** (Liao et al., 2023) | rWL | **7.13** | 8.11 | 9.50 | 2.97 | 2.07 | 4.91 | 2.83 |
| | WNS | -1.33 | -1.28 | -4.71 | -1.12 | -0.82 | -0.73 | 4.67 |
| | TNS | -3640.33 | -3517.62 | -129.52 | -230.09 | -88.98 | -698.98 | 3.67 |
| | DRC | 18 | 26 | 274 | 269 | 79 | 2871 | 3.17 |
| | Power | 0.4537 | 0.5182 | **0.4954** | 0.2025 | 0.2141 | 0.2701 | 4.00 |
| | Cell Area | **38.55** | 40.04 | **52.72** | 12.27 | 7.50 | 23.08 | 2.67 |
| **Hier-RTLMP** (Kahng et al., 2023) | rWL | 7.59 | 7.85 | 9.81 | 3.06 | 2.24 | 5.50 | 3.83 |
| | WNS | -1.13 | -1.19 | **-4.63** | -1.24 | -0.52 | -0.72 | 3.33 |
| | TNS | -2892.99 | -3085.98 | -918.45 | -382.62 | -113.74 | -649.06 | 5.00 |
| | DRC | **6** | 10 | 458 | 2400 | 273 | 4841 | 3.50 |
| | Power | 0.4481 | 0.4991 | 0.4985 | 0.2055 | 0.2130 | 0.2670 | 3.33 |
| | Cell Area | 38.88 | 40.06 | 53.35 | 12.34 | 7.43 | 23.31 | 4.00 |
| **MaskPlace** (Lai et al., 2022) | rWL | 7.21 | **7.80** | 10.94 | **2.63** | 2.21 | 5.52 | 3.33 |
| | WNS | -1.28 | -1.32 | -4.66 | **-1.06** | -0.53 | -1.43 | 4.33 |
| | TNS | -3760.32 | -3713.91 | -142.35 | **-226.37** | -90.62 | -1410.02 | 4.33 |
| | DRC | 20 | 131 | 13435 | 100 | 1255 | 1131 | 4.00 |
| | Power | 0.4512 | 0.4976 | 0.4957 | 0.2032 | 0.2152 | 0.2732 | 3.83 |
| | Cell Area | 38.58 | **39.81** | 53.84 | 12.24 | 7.41 | 23.46 | 3.33 |
| **MaskRegulate** (Xue et al., 2024) | rWL | 8.57 | 8.83 | 10.17 | 3.40 | 2.26 | 5.25 | 5.33 |
| | WNS | -1.06 | -1.14 | -4.76 | -1.16 | **-0.49** | -0.66 | 2.67 |
| | TNS | -2877.32 | -3003.30 | -56.41 | -262.01 | -90.79 | -506.76 | 2.83 |
| | DRC | 9 | 54 | 74 | 9011 | 5581 | 2980 | 4.33 |
| | Power | 0.4494 | 0.5049 | 0.5501 | 0.2079 | 0.2120 | 0.2747 | 5.00 |
| | Cell Area | 39.69 | 40.80 | 53.80 | 12.52 | 7.29 | 23.25 | 4.67 |
| **LaMPlace** (Geng et al., 2025) | rWL | 7.96 | 8.10 | 10.55 | 2.91 | 2.13 | **4.39** | 3.50 |
| | WNS | **-1.04** | -1.18 | -4.84 | -1.36 | -0.53 | -0.76 | 4.17 |
| | TNS | -2758.19 | -3030.05 | -329.68 | -360.87 | -97.90 | -547.59 | 3.83 |
| | DRC | 602 | 9 | 4692 | 395 | 3721 | **1096** | 3.67 |
| | Power | 0.4477 | **0.4969** | 0.5211 | 0.2033 | 0.2095 | **0.2656** | 2.50 |
| | Cell Area | 38.86 | 40.26 | 54.00 | 12.39 | 7.37 | 22.91 | 4.00 |
| **EXPlace (Ours)** | rWL | 7.92 | 7.99 | **8.91** | 2.68 | **1.96** | 4.51 | **2.17** |
| | WNS | -1.07 | **-1.08** | -4.66 | -1.11 | -0.49 | **-0.57** | **1.83** |
| | TNS | **-2711.80** | **-2834.88** | **-50.09** | -257.73 | -77.33 | **-458.80** | **1.33** |
| | DRC | 8 | **8** | 49 | **8** | **0** | 1167 | **1.60** |
| | Power | **0.4472** | 0.4996 | 0.5038 | **0.2012** | 0.2083 | 0.2681 | **2.33** |
| | Cell Area | 39.15 | 40.07 | 52.92 | 12.21 | 7.27 | **22.85** | **2.33** |

(2.17), WNS (1.83), TNS (1.33), DRC (1.60), Power (2.33) and Area (2.33), evidencing consistent superiority. Averaged across designs, EXPlace reduces the magnitude of TNS by 23.06% relative to the analytical placer DREAMPlace and by 5.97% relative to the strongest RL baseline MaskRegulate. EXPlace also delivers the best overall DRC, with notably zero violation on the chip case bp_fe. Compared with the expertise-driven Hier-RTLMP (Kahng et al., 2023), EXPlace attains uniformly better average ranks and yields substantial metric-level gains: rWL improves by 5.77%, WNS by 4.77%, TNS by 20.54% and DRC by 84.48%. These results highlight the advantage of RL with integrated expert priors over strong heuristics, establishing EXPlace as state of the art among both RL-based and traditional expert-driven methods on OpenROAD.

**Timing-driven Fine-tuning.** We perform DPO-based fine-tuning for 25 iterations, with 20 timing evaluations per iteration. At each iteration, we sample multiple placements and evaluate them by OpenTimer (Huang et al., 2020) to obtain *post-placement* TNS and WNS. Since candidates can be non-dominating across these two objectives, we summarize timing quality via the Pareto front (Zhou et al., 2019). Fig. 5 visualizes the Pareto fronts from three stages: the pre-trained policy, after 5 DPO iterations, and after all 25 iterations. Results show that the proposed fine-tuning strategy consistently shifts the Pareto front toward improved timing metrics. On superblue1, for instance, TNS improves from -1070.14 to -821.20 and WNS from -51.48 to -36.48, yielding a significant average improvement ratio of 26.20%. These data indicate that the DPO-based fine-tuning approach can effectively leverage timing preferences to jointly optimize TNS and WNS. Results on other 5 cases are provided in App. D.5. We further assess alignment between the policy's likelihoods and timing preferences. For each trajectory pair, we label it as consistent if the trajectory with better TNS and WNS is assigned higher likelihood by the policy. The right-bottom subfigure of Fig. 5 reports the percentage of consistent pairs. Fine-tuning clearly increases this consistency relative to the pre-trained policy, suggesting that the learned solution distribution becomes better aligned

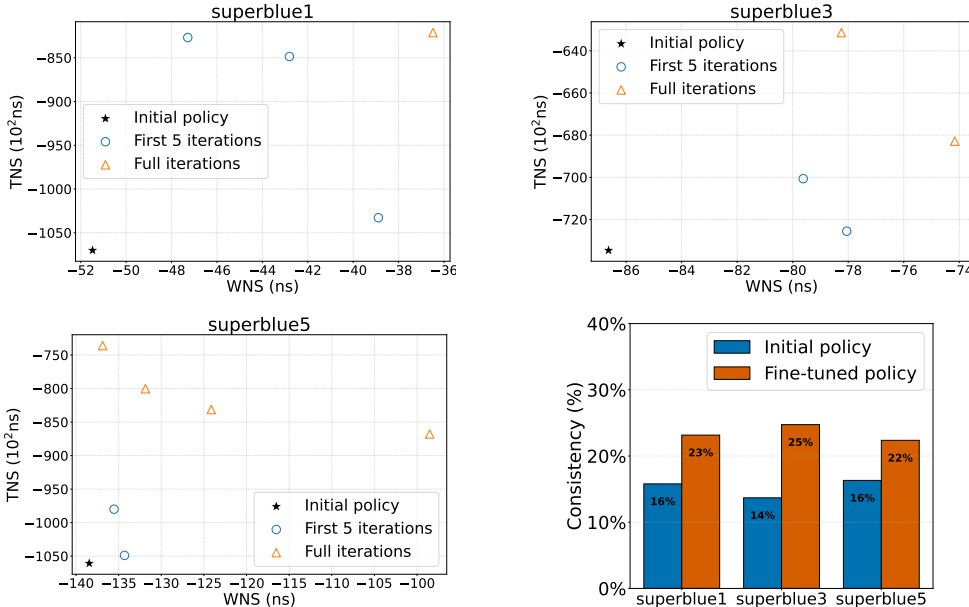

Figure 5: Results of timing-driven fine-tuning on ICCAD 2015 benchmark. The first three subfigures visualize Pareto optimal points of TNS and WNS at different optimization stages, while the right-bottom subfigure presents the comparison between the initial policy and the fine-tuned policy, on the consistency (the larger the better) between the policy's likelihoods and timing preferences.

with timing-oriented ground truth. Finally, we want to emphasize that this timing-driven fine-tuning process is generally time-consuming, requiring about 15-30 hours on ICCAD 2015 cases due to repeated timing evaluations. But in real-world scenarios where the design cycle can accommodate such costly iterative updates, this fine-tuning strategy is an effective choice for timing optimization.

**Additional Results.** Due to space limitation of the main paper, we present additional results in App. D, including the ablation study of each expertise component (App. D.2), generalization study of RL-trained policies (App. D.3), and runtime analysis of EXPlace (App. D.1). These results further validate the effectiveness of our proposed modules. To facilitate a more intuitive comparison, we also present a series of visualization results in App. D.6, where it can be observed that EXPlace exhibits more regular and human expert-level placement layouts.

## 6 CONCLUSIONS

This work presents EXPlace, an expertise-augmented learning-based framework that makes RL placer more practical and effective for real chip design flows. Motivated by the limitations of prior RL methods that optimize oversimplified proxy objectives and overlook years of industrial expertise, EXPlace integrates domain expertise from two complementary perspectives. First, *Expert Knowledge Injection* transforms well-established placement expertise, including dataflow guidance, macro grouping, periphery biasing, and I/O keepout, into expressive state representations and dense rewards, aligning step-wise decisions with human design intent. Second, *Expert Workflow Imitation* closes the gap between proxy rewards and final PPA by incorporating timing-driven preference optimization, emulating the iterative refinement loop used by human engineers. Extensive evaluations on ICCAD 2015 and OpenROAD benchmarks demonstrate that EXPlace consistently improves wire-length and timing metrics, while maintaining low DRC violations. Compared to various advanced placers, EXPlace achieves the best average ranks and substantial performance gains, confirming the value of combining expertise with data-driven learning. One limitation of our work is that the cost of timing-driven fine-tuning process. Future works could include improving the timing-driven optimization efficiency, as well as developing fast and accurate timing predictors to replace expensive evaluations of static timing analysis, thereby accelerating the fine-tuning process.

ACKNOWLEDGMENTS

This work was supported by the National Science and Technology Major Project (2022ZD0116600), the Jiangsu Science Foundation Leading-edge Technology Program (BK20232003), the Fundamental Research Funds for the Central Universities (14380020), and the National Science Foundation of China (624B2069). The authors want to acknowledge support from the Huawei Technology Cooperation Project.

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

## A    DETAILED BACKGROUND

Chip planning bridges front-end design and back-end implementation (Tseng, 2024). It determines the physical locations of design components and primarily consists of two steps: floorplanning and placement, which address different inputs and pursue different objectives. **Chip floorplanning** operates at a higher level of abstraction, where the decision variables are blocks. The task is to arrange multiple black-box blocks given their sizes, maximum aspect ratios, and pairwise connection strengths. Its goal is to pack these blocks to minimize both the total inter-block connection cost and the overall bounding-box area (Zhong et al., 2024; Li et al., 2023). Sometimes it also decides the pin locations to implement inter-block connections (Xu et al., 2025; Pedram et al., 1990). **Chip placement** implements a floorplanned block by determining the detailed locations of macros and standard cells within that block. In most well-established flows, it is divided into two stages: macro placement and standard cell placement (Ajayi et al., 2019; Mirhoseini et al., 2021). There are also academic and commercial tools supporting concurrent mixed-size placement (Chen et al., 2023; Patwardhan, 2020); however, final macro decisions are still made cautiously due to their large footprints and special design rules.

### A.1    REINFORCEMENT LEARNING FOR MACRO PLACEMENT

Recently, researchers have applied RL to obtain high-quality macro placement. AlphaChip (Mirhoseini et al., 2021; Goldie et al., 2024) models macro placement as a Markov decision process (MDP) for the first time and proposes a two-stage pipeline: a RL agent places the macros one by one in the first stage, and an analytical placement tool handles the remaining components in the second stage. AlphaChip divides the chip canvas into discrete grids, and then learns to place one macro to a valid grid coordinate per step until all are placed. Early work adopted sparse rewards—zero until all macros are placed—followed by a terminal reward based on global placement wirelength and congestion (Mirhoseini et al., 2021; Cheng & Yan, 2021). While this aligns with the black-box nature of chip optimization, it leads to slow training, unstable convergence, and heavy computational overhead. After that, MaskPlace (Lai et al., 2022) addresses these issues by relying on pixel-level visual features (masks) and optimizing step-wise macro wirelength increments, yielding dense rewards and steadier convergence. Concretely, given a partially placed design, the placement canvas is discretized into grids that serve as the action space. For each grid, the method computes the wirelength increment if the next macro is placed there. This "WireMask" both informs the state representation and provides the wirelength-based reward. Building on the same mask framework, ChiPFormer (Lai et al., 2023) employs an offline decision transformer to improve generalization. Recently, MaskRegulate (Xue et al., 2024) introduces a regularity mask to encourage peripheral placements that reflect expert preferences, followed by FlowPlace (Xie et al., 2026), which further improves the routability of generative placement methods.

However, many RL-based macro placement methods primarily optimize macro half-perimeter wirelength (HPWL) (Lai et al., 2022; 2023; Shi et al., 2023; Xue et al., 2024; Geng et al., 2024) by extracting nets that connect only macros. This objective does not faithfully represent the placement objectives. Although these methods often reflect global placement wirelength, the underlying optimization target can be misleading (Wang et al., 2024): it may cluster macros excessively and harm the overall circuit metrics that actually matter. In the next section, we discuss additional expert knowledge that helps optimize PPA performance beyond this basic objective.

### A.2    CHIP PLACEMENT METHODS THAT INCORPORATE EXPERT KNOWLEDGE

Human engineers rely on expertise honed through thousands of design cycles to quickly narrow down high-quality placement candidates. For instance, they group macros that serve the same logical function to reduce wirelength and simplify power planning; they place macros at chip peripheries to avoid blocking core-region routing resources, which is critical for signal integrity; they reserve I/O port regions to ensure buffering for long-distance signals, preventing timing repair failures. Recently, there are several efforts that attempt to incorporate expert knowledge into automatic macro placers, as shown in Table 1.

Among analytical placers, IncreMacro (Pu et al., 2024) designs a smoothed and differentiable periphery cost to push macros to the chip periphery. However, many other forms of expertise are

difficult to be differentiable, and the learnability and generalizability of analytical placers are not as good as those of RL methods. Several BBO-based placers incorporate dataflow into placement (Vidal-Obiols et al., 2019; Lin et al., 2021; Zhao et al., 2025), which capture the hidden connections and better reveal the design intentions of the frontend engineers. Lin et al. (2019a) and Kahng et al. (2022) used macro grouping that regularly place macros that belong to the same hierarchy and share the same footprint, packing them to improve the regularity of the placement. Hier-RTLMP (Kahng et al., 2023) consists of a wide range of expertise-driven cost functions, including dataflow, notch avoidance, periphery placement, I/O keepout, and dead space. The final BBO objective is the weighted sum of these cost functions. However, these optimization-based approaches are constrained by their non-learnable nature, making them unable to utilize real backend PPA feedback for iterative refinement, ultimately leading to limited improvements in final PPA performance.

Few learning-based macro placers consider expertise. Le et al. (2023) proposed to place macros to the candidate corners, and incorporates macro grouping information into both node embedding and reward function. This work does provide various expertise to consider, but it builds upon the GraphPlace and faces sparse reward issue. Xue et al. (2024) turned to adopt MaskPlace's vision solution (Lai et al., 2022) to provide more direct features and dense reward to stabilize training procedure, and proposed a regularity mask to push macros to the periphery. It shows promising performance, while it contains only peripheral placement expertise. We try to extend to three more masks containing dataflow, macro grouping, and I/O region keepout, to explicitly guide the search of RL agent. LaMPlace (Geng et al., 2025) is the only one capable of using real backend PPA to enhance the placement performance. However, it is highly data-dependent, coupled with the inherent scarcity and confidentiality of industrial chip data, making its real-world performance and robustness unproven, particularly in the absence of human expertise guidance.

## B DETAILS OF EXPLACE

We provide the detailed transition and policy training of EXPlace in this section.

**Transition.** At time $t$, the agent selects a valid grid $a_t = (x, y)$ for placing the next macro in a pre-defined macro ordering (Lai et al., 2022) that prioritizes macros with high connectivity density and large footprints. Upon committing $a_t$, the partial layout is updated, and all feasibility and expert masks are re-computed accordingly. The episode terminates once all macros have been placed.

**Policy Network and Training.** The canvas and masks of state $s_t$ are concatenated as a multi-channel visual input and processed by a CNN with local and global branches (Lai et al., 2022). The local branch uses small kernels to fuse input masks, while the global branch extracts high-level features and upsamples them via transposed convolutions. A linear layer merges both branches to produce $\pi_\theta(s_t)$. We train the policy with PPO (Schulman et al., 2017), using an MLP critic over global features to estimate the state-value function.

## C DETAILS OF EXPERIMENTS

**Hyperparameters.** The trade-off weights for the rewards are summarized in Table 4. In general, we assign higher weights to periphery biasing and macro grouping to encourage regular placements, while down-weighting the I/O keepout reward to preserve flexibility. The dataflow and HPWL rewards receive equal weight. The only difference between the two benchmark configurations is an increased grouping weight for the OpenROAD cases, which exhibit clear hierarchy and uniformly sized macros, thereby amplifying the inherent bias toward grouping. Most RL training and network architecture hyperparameters follow the default settings of MaskPlace (Lai et al., 2022): 1,500 training episodes, 10 training epochs, and a buffer size equal to 10 times the trajectory length. The only change is a larger batch size of 256 to increase device utilization and improve training efficiency. For timing-driven fine-tuning, we set the TNS/WNS threshold to the top 10% of pairwise TNS/WNS differences, i.e., we retain only the most distinguishable preference pairs to reduce noise. The DPO training uses a batch size of 20, with 5 epochs per iteration.

**Implementation Details.** Unlike previous RL methods, we implement parallel environments so that multiple trajectories can be sampled simultaneously, improving the efficiency of both RL and

Table 4: Weight configuration on the ICCAD 2015 contest and OpenROAD benchmarks.

| Weight | Configuration on ICCAD 2015 | Configuration on OpenROAD |
|---|---|---|
| Dataflow weight $\lambda_0$ | 0.15 | 0.15 |
| Grouping weight $\lambda_1$ | 0.2 | 0.3 |
| Periphery weight $\lambda_2$ | 0.45 | 0.35 |
| I/O keepout weight $\lambda_3$ | 0.05 | 0.05 |
| HPWL weight $\lambda_4$ | 0.15 | 0.15 |

DPO. The number of parallel environments is set to match the buffer size exactly, so a single parallel rollout fills the buffer. On the ICCAD 2015 contest benchmark, we observed many small components that could be misclassified as macros. To address this, we apply an area-based filter to select only components with sufficiently large footprints as macros. For fairness, the same macro-selection configuration is used across all baselines. For dataflow graph construction, we use breadth-first search to find macro-to-macro paths. To avoid long paths and overly dense connections, we cap the hop count at 2; that is, paths traversing more than two flip-flop nodes are excluded from the graph.

# D   ADDITIONAL RESULTS

## D.1   RUNTIME ANALYSIS

The mask runtime breakdown in Fig. 6 shows that the regularity and dataflow masks account for the majority of mask computation. Because the dataflow graph is much denser than the HPWL connectivity (see App. D.4), computing the dataflow mask incurs substantially higher runtime than computing the wire mask. Nevertheless, the training-time comparison in Table 5 indicates that the training of EXPlace is faster than the original implementation of MaskRegulate (Xue et al., 2024). This is largely due to that we parallelize the environment transition computation across multiple episodes, which significantly reduces mask-computation overhead. Moreover, given the overall placement quality improvements (as shown in Table 2 and 7), the additional runtime from expert masks is a worthwhile trade-off for better final circuit performance.

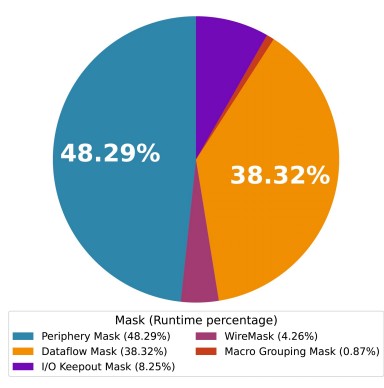

Figure 6: Mask runtime breakdown.

Compared to BBO and analytical methods, although the case-by-case training of EXPlace is time-consuming, its inference speed is significantly faster, e.g., 5.28 seconds on superblue1, which is orders of magnitude faster than the 287s of the analytical DREAMPlace (Liao et al., 2023) and the 792s of the BBO-based LaMPlace (Geng et al., 2025). In App. D.3, we further compare the zero-shot inference performance of EXPlace with the full-optimization performance of BBO and analytical methods. The results demonstrate that the zero-shot EXPlace still outperforms both approaches while maintaining a lightweight inference cost.

Table 5: Runtime (in seconds) of various methods on the ICCAD 2015 benchmark. The runtime of LaMPlace is reported as in the original paper.

| Method | Time Type | superblue1 | superblue3 | superblue4 | superblue5 | superblue7 | superblue10 | superblue16 | superblue18 | Avg. |
|---|---|---|---|---|---|---|---|---|---|---|
| DREAMPlace | Test | 287.30 | 158.11 | 107.54 | 121.20 | 199.03 | 210.90 | 112.70 | 66.12 | 157.86 |
| LaMPlace | Test | 792.00 | 1368.00 | 396.00 | 2592.00 | 756.00 | 11376.00 | 36.00 | 180.00 | 2187.00 |
| MaskPlace | Train | 3376.95 | 4755.24 | 6369.17 | 4514.60 | 11975.79 | 4949.54 | 2623.18 | 2571.54 | 5142.00 |
| | Test | 1.64 | 2.70 | 3.88 | 2.28 | 5.93 | 3.46 | 2.59 | 3.11 | 3.20 |
| MaskRegulate | Train | 8534.70 | 12636.81 | 15743.53 | 11497.57 | 30039.77 | 10109.45 | 6122.82 | 6381.81 | 12633.31 |
| | Test | 4.69 | 8.08 | 8.92 | 6.89 | 17.95 | 8.09 | 3.37 | 3.72 | 7.72 |
| EXPlace (Ours) | Train | 6984.49 | 8384.76 | 11036.91 | 7723.30 | 15939.91 | 8696.94 | 5178.73 | 5051.55 | 8624.57 |
| | Test | 5.28 | 9.31 | 10.68 | 6.80 | 19.55 | 7.49 | 5.03 | 4.97 | 8.64 |

## D.2 ABLATION STUDY

We conduct ablation studies on dataflow guidance, macro grouping, periphery bias, and I/O keepout across three representative ICCAD 2015 cases. As shown in Table 6, each component contributes positively, and the full EXPlace achieves the best overall performance.

Among the four expertise components, dataflow guidance and periphery biasing exert the most pronounced influence. Removing dataflow guidance leads to substantial degradation in timing metrics (WNS and TNS), which is expected since dataflow paths closely correlate with the critical timing paths that span between macros and flip-flops; the absence of such guidance thus directly impairs timing closure. Without periphery biasing, both routing overflow (rOverflowH and rOverflowV) and routed wirelength (rWL) deteriorate markedly, as macros placed in the core region obstruct lower metal layers used for signal routing, intensifying congestion and forcing lengthy detours. This confirms that peripheral placement is essential for maintaining routability and achieving competitive wirelength. Macro grouping also contributes meaningfully, particularly to wirelength and timing. These ablation results collectively validate the necessity of each expertise component and the rationale behind our reward weighting strategy.

Table 6: Ablation study on three representative cases. "w/o" denotes "without". We ablate by removing, one at a time, the I/O port keepout (w/o I/O Port Keepout), dataflow guidance (w/o Dataflow), macro grouping (w/o Grouping), and periphery bias (w/o Periphery) components.

| Method | Metric | superblue1 | superblue3 | superblue4 | Avg. Rank |
|---|---|---|---|---|---|
| **EXPlace (w/o Dataflow)** | rWL | 114 | 144 | 80.70 | 3.33 |
| | rOverflowH | 0.05% | **0.88%** | **0.17%** | 1.33 |
| | rOverflowV | 0.03% | **0.05%** | 0.03% | 1.67 |
| | NVP | **10.90** | 11.10 | 9.76 | 3.00 |
| | WNS | -65.30 | -100.90 | -40.70 | 4.00 |
| | TNS | -9.88 | -14.20 | -9.76 | 4.00 |
| **EXPlace (w/o Grouping)** | rWL | 115 | 138 | 85.50 | 3.67 |
| | rOverflowH | 0.16% | 0.88% | 0.21% | 2.67 |
| | rOverflowV | 0.03% | 0.09% | **0.02%** | 2.33 |
| | NVP | 15.70 | 10.00 | 9.13 | 3.33 |
| | WNS | **-50.50** | **-78.80** | -51.20 | 2.00 |
| | TNS | -10.90 | -10.10 | -9.22 | 3.67 |
| **EXPlace (w/o I/O Port Keepout)** | rWL | 115 | 136 | 79.40 | 2.67 |
| | rOverflowH | **0.03%** | 1.27% | 0.47% | 3.00 |
| | rOverflowV | 1.27% | 0.07% | 0.04% | 3.33 |
| | NVP | 13.80 | **1.10** | 9.45 | 2.33 |
| | WNS | -54.20 | -100.00 | -54.00 | 4.00 |
| | TNS | -9.69 | -9.09 | -9.09 | 2.33 |
| **EXPlace (w/o Periphery)** | rWL | 121 | 155 | 80.60 | 4.33 |
| | rOverflowH | 0.87% | 6.55% | 1.42% | 5.00 |
| | rOverflowV | 0.21% | 0.41% | 0.15% | 4.67 |
| | NVP | 14.00 | 15.70 | 9.92 | 4.67 |
| | WNS | -51.90 | -82.10 | -44.90 | 2.33 |
| | TNS | -11.80 | -14.50 | **-6.24** | 3.67 |
| **Ours** | rWL | **108** | **134** | **77.50** | 1.00 |
| | rOverflowH | 0.15% | 1.24% | 0.27% | 3.00 |
| | rOverflowV | **0.02%** | 0.10% | 0.05% | 3.00 |
| | NVP | 13.20 | 9.86 | **8.77** | 1.67 |
| | WNS | -58.60 | -82.30 | **-34.10** | 2.67 |
| | TNS | **-8.63** | **-7.80** | -6.65 | 1.33 |

### D.3 GENERALIZATION RESULTS

Generalization across different cases is important to evaluate RL-based placers. To evaluate the generalization performance, we directly apply the model trained on **superblue1** to solve the macro placement of other cases. Under this challenging generalization scenario (as presented in Table 7), EXPlace consistently achieves superior average performance across key metrics, including rWL, NVP, and TNS, surpassing all RL-based, BBO-based, and analytical baselines. In particular, when compared to the analytical approach DREAMPlace (Liao et al., 2023), EXPlace demonstrates significantly reduced routing overflow, attributable to its enhanced placement regularity. This improvement in placement quality subsequently leads to better routed wirelength and timing performance.

Furthermore, the inference time of EXPlace is efficient. For example, on superblue1, it requires only 5.28 seconds, which represents a substantial speedup compared to the 287 seconds required by the analytical DREAMPlace (Liao et al., 2023) and the 792 seconds by the BBO-based LaMPlace (Geng et al., 2025). Despite the lower runtime of MaskPlace (Lai et al., 2022) and MaskRegulate Xue et al. (2024), the consistently superior performance of EXPlace warrants the moderate increase in runtime.

Table 7: Generalization results of RL-based placement methods on ICCAD2015 benchmarks. All the RL models are trained on superblue1 and directly generalized to other cases. Units: rWL (m), rOverflowH (%), rOverflowV (%), NVP ($10^3$), WNS (ns), TNS ($10^4$ ns), Runtime (s).

| Method | Metric | superblue1 | superblue3 | superblue4 | superblue5 | superblue7 | superblue10 | superblue16 | superblue18 | Avg. Rank |
|---|---|---|---|---|---|---|---|---|---|---|
| **DREAMPlace** (Liao et al., 2023) | rWL | 159 | 199 | 140 | 167 | 220 | 244 | 136 | **53.3** | 4.50 |
| | rOverflowH | 16.93% | 15.05% | 18.74% | 5.67% | 6.93% | 3.83% | 16.37% | 0.19% | 4.75 |
| | rOverflowV | 4.72% | 7.40% | 11.60% | 2.18% | 5.24% | 2.10% | 1.93% | 0.02% | 4.75 |
| | NVP | 30.0 | 19.8 | 26.9 | 20.5 | 38.1 | 24.5 | 37.5 | **5.02** | 4.50 |
| | WNS | -111 | -108 | -106 | **-71.7** | -59.6 | -240 | -65.5 | -30.3 | 3.75 |
| | TNS | -37.3 | -25.1 | -21.8 | -17.4 | -13.7 | -41.0 | -21.0 | -1.78 | 4.25 |
| | Runtime | 287.30 | 158.11 | 107.54 | 121.20 | 199.03 | 210.90 | 112.70 | 66.12 | 4.12 |
| **LaMPlace** (Geng et al., 2025) | rWL | 117 | 103.1 | 95.3 | **131** | 189 | 210 | 100 | 58.3 | 2.25 |
| | rOverflowH | 0.19% | **0.02%** | 1.50% | 0.99% | 0.16% | 0.06% | 0.94% | 0.19% | 2.50 |
| | rOverflowV | 0.07% | **0.01%** | 0.10% | 0.15% | 0.04% | 0.03% | 0.04% | 0.02% | 2.62 |
| | NVP | 14.3 | **4.24** | 13.6 | 18.1 | **18.1** | 16.7 | 18.0 | 7.48 | 2.38 |
| | WNS | **-47.2** | **-76.2** | -46.6 | -117 | -59.8 | -69.9 | -37.2 | -29.6 | **2.12** |
| | TNS | -12.9 | **-3.49** | -10.7 | -10.8 | -12.0 | -33.0 | -15.3 | -3.60 | 2.12 |
| | Runtime | 792.00 | 1368.00 | 396.00 | 2592.00 | 756.00 | 11376.00 | 36.00 | 180.00 | 4.88 |
| **MaskPlace** (Lai et al., 2022) | rWL | 140 | 167 | 104 | 158 | 200 | 203 | 107 | 66.9 | 3.88 |
| | rOverflowH | 6.59% | 3.85% | 5.46% | 1.05% | 1.00% | 0.20% | 5.26% | 0.35% | 4.12 |
| | rOverflowV | 0.46% | 0.33% | 0.32% | 0.19% | 0.17% | 0.06% | 0.08% | 0.22% | 4.00 |
| | NVP | 29.5 | 17.3 | 20.9 | 17.2 | 29.5 | 19.2 | 21.0 | 7.85 | 4.00 |
| | WNS | -50.1 | -121 | -64.8 | -164 | -51.0 | **-55.6** | -44.0 | -30.2 | 3.00 |
| | TNS | -22.0 | -18.5 | -15.1 | -18.8 | -13.9 | -28.1 | -17.8 | -5.18 | 3.88 |
| | Runtime | 1.64 | 2.70 | 3.88 | 2.28 | 5.93 | 3.46 | 2.59 | 3.11 | 1.00 |
| **MaskRegulate** (Xue et al., 2024) | rWL | 118 | 153 | 87.1 | 141 | 186 | 237 | 104 | 59.9 | 3.00 |
| | rOverflowH | **0.06%** | 0.31% | **0.06%** | 0.03% | **0.02%** | 0.10% | **0.18%** | 0.01% | 1.38 |
| | rOverflowV | **0.02%** | 0.04% | **0.02%** | 0.01% | 0.01% | 0.06% | 0.03% | 0.00% | 1.50 |
| | NVP | **11.6** | 12.0 | 10.1 | 13.4 | 19.1 | 17.8 | 18.1 | 6.12 | 2.25 |
| | WNS | -66.1 | -114 | -57.3 | -133 | -66.2 | -62.3 | **-34.1** | -37.5 | 3.38 |
| | TNS | -13.2 | -15.2 | -12.1 | -11.0 | -13.9 | -36.7 | **-13.8** | -2.18 | 3.00 |
| | Runtime | 4.69 | 8.08 | 8.92 | 6.89 | 17.95 | 8.09 | 3.37 | 3.72 | 2.25 |
| **EXPlace (Ours)** | rWL | **111** | 141 | 79.6 | 133 | 179 | 174 | 96.0 | 55.8 | **1.38** |
| | rOverflowH | 0.07% | 0.65% | 0.47% | 0.10% | 0.35% | **0.03%** | 0.69% | 0.08% | 2.12 |
| | rOverflowV | 0.02% | 0.05% | 0.02% | 0.02% | 0.03% | **0.00%** | 0.02% | 0.01% | 1.62 |
| | NVP | 12.5 | 10.4 | **9.94** | **11.5** | 19.3 | 16.3 | 18.8 | 5.42 | **1.88** |
| | WNS | -54.6 | -92.9 | **-40.6** | -121 | -68.8 | -66.8 | -47.8 | **-16.0** | 2.75 |
| | TNS | **-9.93** | -10.8 | **-8.85** | **-9.87** | -13.0 | -24.1 | -18.8 | -1.24 | **1.62** |
| | Runtime | 5.28 | 9.31 | 10.68 | 6.80 | 19.55 | 7.49 | 5.03 | 4.97 | 2.75 |

### D.4 COMPARISON OF DATAFLOW AND HPWL CONNECTIONS

Table 8 reports the connection densities of HPWL (i.e., the original netlist) and the dataflow graph. It shows that the dataflow graph is orders of magnitude denser, providing a more detailed and comprehensive representation of macro-to-macro relationships. Intuitively, the dataflow graph incorporates indirect connections between macros mediated by numerous standard cells, thereby capturing how macro placement influences standard-cell distribution and enabling standard cell-aware macro placement. Therefore, incorporating the dataflow guidance into macro placement is essential for achieving decent PPA metrics.

Table 8: Density statistics of macro HPWL and dataflow on the ICCAD 2015 contest benchmark.

| case | Density of HPWL | Density of dataflow |
|---|---|---|
| superblue1 | 1.77% | 17.20% |
| superblue3 | 0.94% | 10.35% |
| superblue4 | 1.48% | 11.62% |
| superblue5 | 0.08% | 50.73% |
| superblue7 | 2.04% | 6.01% |
| superblue10 | 2.11% | 26.89% |
| superblue16 | 1.42% | 19.88% |
| superblue18 | 1.48% | 18.41% |

### D.5 COMPARISON WITH AND WITHOUT TIMING-DRIVEN FINE-TUNING

We compare the performance of the proposed EXPlace with and without timing-driven fine-tuning (i.e., DPO). The results are presented in Table 9. It can be observed that the timing-driven fine-tuning process significantly improves TNS by 12.29% and WNS by 22.68% on average, while the reduction in global HPWL is marginal. These results indicate that DPO effectively optimizes critical timing paths to achieve better TNS and WNS, rather than merely reducing HPWL.

Table 9: Comparison of EXPlace with and without timing-driven fine-tuning (i.e., DPO) on ICCAD 2015 contest benchmarks. The evaluation metrics include Global HPWL (m), TNS ($\times 10^2$ ns), WNS (ns). The best results are highlighted in **bold**.

| Design | EXPlace | | | EXPlace (DPO) | | |
|---|---|---|---|---|---|---|
| | TNS | WNS | HPWL | TNS | WNS | HPWL |
| superblue1 | -1070.14 | -51.48 | **534.00** | **-821.20** | **-36.48** | 559.23 |
| superblue3 | -734.65 | -86.83 | 654.10 | **-682.85** | **-74.16** | **637.13** |
| superblue4 | -668.98 | -39.48 | 382.10 | **-474.97** | **-22.55** | **377.35** |
| superblue5 | -1060.71 | -138.45 | 690.30 | **-867.83** | **-98.52** | **650.10** |
| superblue7 | -1227.42 | -61.34 | 832.90 | **-1045.91** | **-47.33** | **810.48** |
| superblue10 | **-2168.76** | -96.34 | 969.70 | -2499.09 | **-65.08** | **892.20** |
| superblue16 | -339.35 | -17.62 | **449.10** | **-284.59** | **-17.09** | 449.97 |
| superblue18 | -106.30 | -21.69 | **269.90** | **-100.81** | **-20.01** | 273.24 |
| Avg. Improvement | – | – | – | **12.29%** | **22.68%** | **1.77%** |

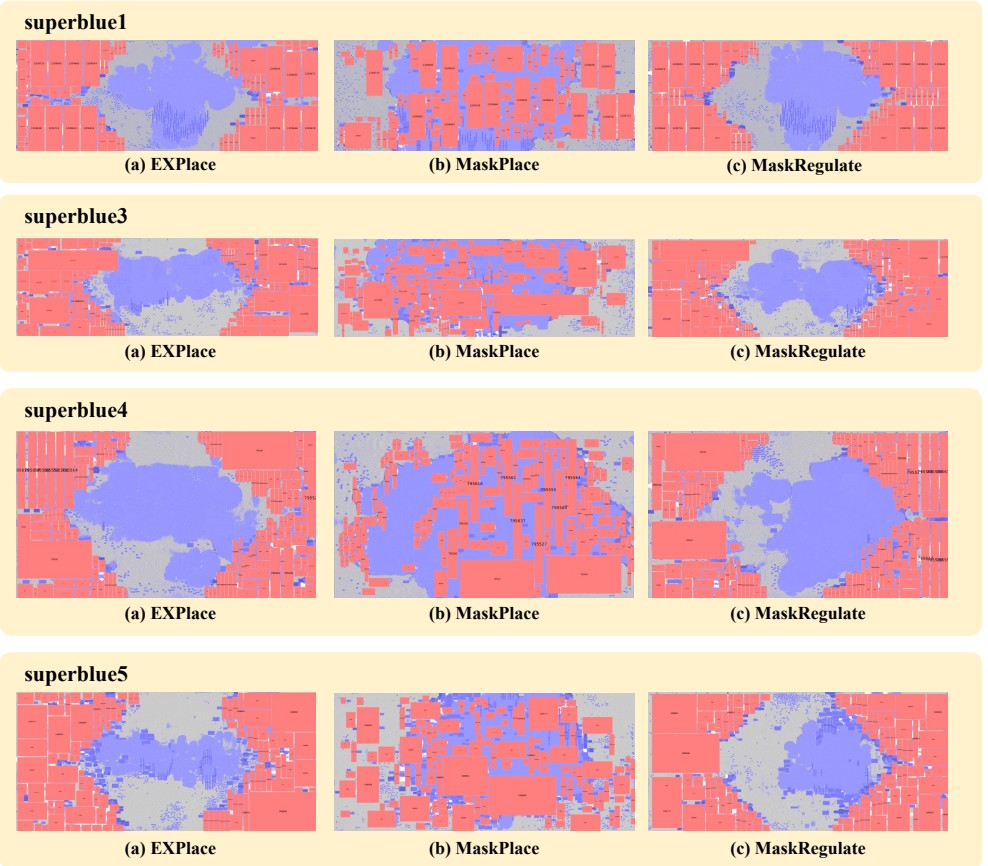

Figure 7: Post-global placement stage visualization of different methods on the ICCAD 2015 benchmark. (a) Our proposed EXPlace. (b) MaskPlace (Lai et al., 2022). (c) MaskRegulate (Xue et al., 2024). Red rectangles represent the macros placed by each method, respectively, and the blue ones represent the smaller macros and cells placed by DREAMPlace.

## D.6 VISUALIZATION OF PLACEMENT RESULTS

**ICCAD 2015 Contest Benchmark.**   We visualize the first four cases superblue1, 3, 4, 5 in Fig. 7 to compare EXPlace with two representative RL-based methods, MaskPlace (Lai et al., 2022) and MaskRegulate (Xue et al., 2024). Compared with MaskPlace, EXPlace tends to push macros toward the chip periphery, creating a continuous region that facilitates standard cell optimization. A notable distinction from MaskRegulate lies in placement regularity: MaskRegulate tends to stack macros in a size-descending order from the periphery inward, interleaving macros from different functional modules, whereas EXPlace groups macros of similar size and hierarchical affinity into cohesive clusters. This regular arrangement better leverages the hierarchical structure of the netlist, benefiting power grid routing and reducing routing congestion. Meanwhile, MaskRegulate's dense size-graded stacking can also prevent cells from being placed between macros, potentially degrading wirelength performance. We further observe that some smaller macros in EXPlace are placed near the chip boundary rather than being grouped with interior clusters. We attribute this to the influence of I/O port keepout regions: the additional whitespace near the periphery naturally accommodates small macros instead of large ones. We also note that our placements do not strictly adhere to macro tiling constraints. For example, on the left side of superblue3, large macros are surrounded by smaller ones. This pattern likely results from the incorporation of dataflow: smaller macros exhibiting stronger communication intensity with the large macros are co-placed nearby.

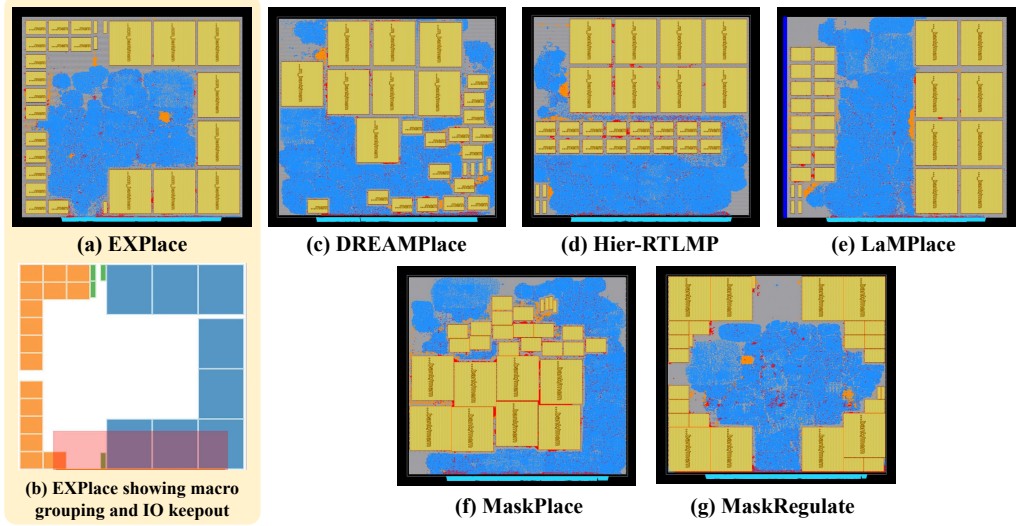

Figure 8: Post-routing stage visualization of different methods on the chip case swerv_wrapper. (a) Our proposed EXPlace. (b) Visualization of the macro grouping and I/O keepout, where macros in the same color belong to an identical group and the red shading is the I/O keepout region. (c-g) Other baselines.

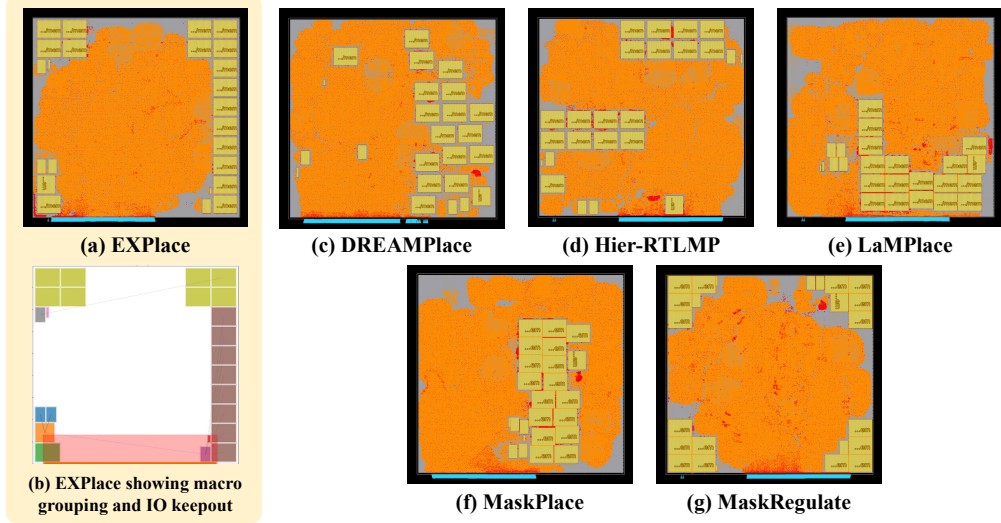

Figure 9: Post-routing stage visualization of different methods on the chip case bp. (a) Our proposed EXPlace. (b) Visualization of the macro grouping and I/O keepout, where macros in the same color belong to an identical group and the red shading is the I/O keepout region. (c-g) Other baselines.

**OpenROAD Benchmark.** We complete the full OpenROAD-flow-scripts (Ajayi et al., 2019) backend implementation, and present final layout visualizations compared with DREAMPlace (Liao et al., 2023), Hier-RTLMP (Kahng et al., 2023), LaMPlace (Geng et al., 2025), MaskPlace (Lai et al., 2022), and MaskRegulate (Xue et al., 2024) in Fig. 8 and 9. We also provide macro grouping and I/O keepout details during optimization. We can clearly observe that the proposed EXPlace effectively captures the intent to group specific macros while maintaining peripheral placement. A large contiguous white space is reserved for standard cell placement and is well-utilized by cells, whereas inevitable detours appear in DREAMPlace, Hier-RTLMP, MaskPlace, and LaMPlace. Although

MaskRegulate introduces a certain degree of regularity, it tends to push macros into the corners, introducing notch areas in the middle parts of the boundary. EXPlace also learns to leave white space around I/O ports, but not strictly, as validated by the reduced DRC violations shown in Table 3.

## D.7 REWARD CURVES DURING TRAINING

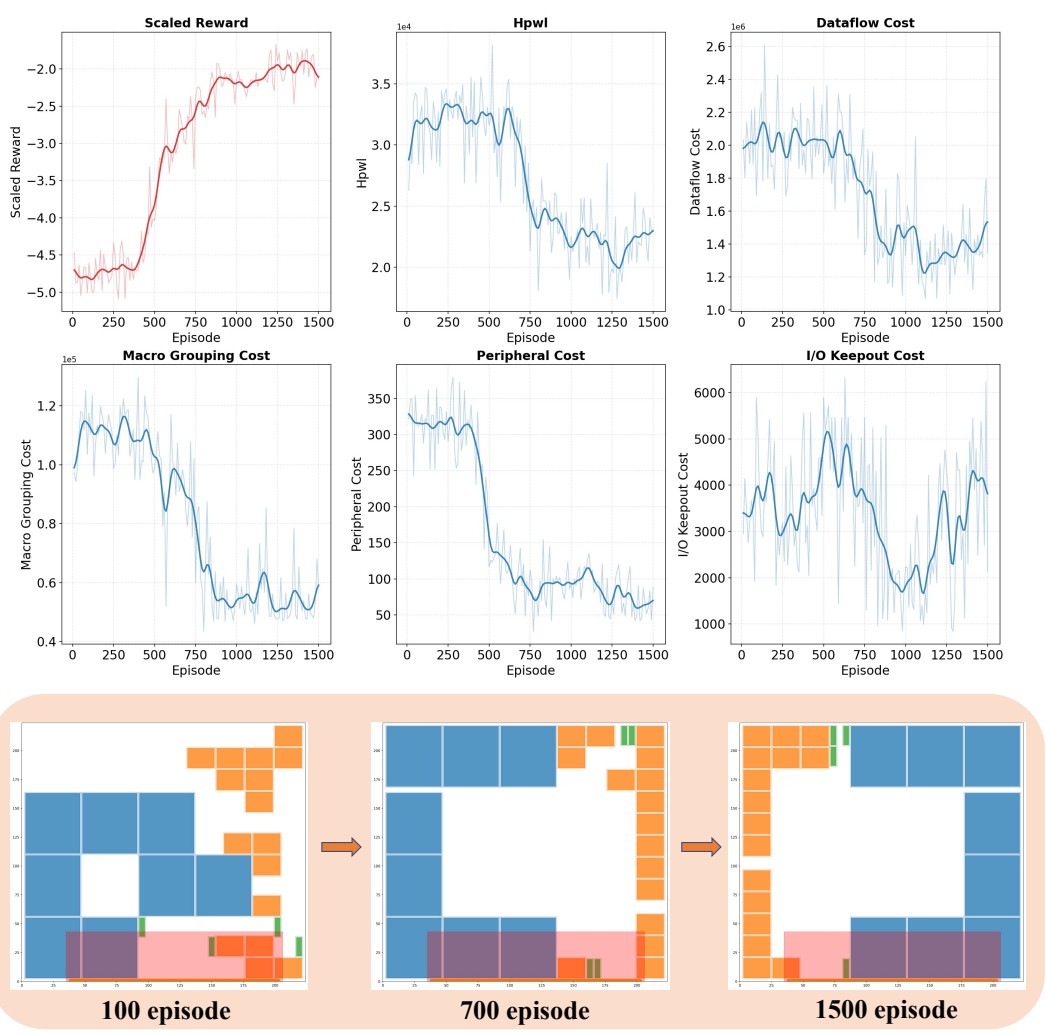

Figure 10: Top: Reward and cost function curves during the training process on swerv_wrapper. Bottom: Visualization of the macro placement evolution throughout training. Macros in the same color belong to an identical group and the red shading is the I/O keepout region.

In Fig. 10, we present the reward and cost function curves during the training process, along with the evolution of macro placement over time. It can be observed that the total scaled reward converges in a stable manner, and the costs associated with HPWL, dataflow, macro grouping, and peripheral biasing all decrease, demonstrating the effectiveness of our reinforcement learning training. The only exception is the I/O keepout cost, which exhibits fluctuations during training. This behavior may result from the relatively low weight assigned to the I/O keepout term to encourage flexible placement, as well as its intrinsic conflict with peripheral biasing—given that I/O ports are distributed along the periphery. Consequently, improvements in peripheral biasing may negatively impact the I/O keepout cost. Despite the fluctuations in I/O keepout cost, the bottom visualization indicates that the policy does learn to avoid I/O ports by placing more macros on the top side of the layout, rather than the bottom, which is densely populated with I/O ports.

# E    THE USAGE OF LARGE LANGUAGE MODELS

We used large language models (LLMs), specifically GPT-5 and GPT-4o, solely to assist in polishing and refining the writing. The LLM was used to rephrase sentences to improve clarity and fluency, and to check for grammatical errors. All ideas, experimental design, results, analysis, and conclusions are entirely produced by ourselves. The LLM was not used to generate any core intellectual content, and we take full responsibility for the entire work.

