# OpenReview forum: "Expertise Can Be Helpful for Reinforcement Learning-based Macro Placement"
_ICLR.cc/2026/Conference — ICLR 2026 Poster_

### Official Review · Reviewer_zEBD · 2025-10-20

**Soundness:** 3
**Presentation:** 3
**Contribution:** 2
**Rating:** 4
**Confidence:** 3

**Summary:**

This paper proposes the EXPlace algorithm to addresse the macro placement task. To mitigate the issue of the algorithm deviating significantly from expert-designed solutions, they employ two key strategies: Expert Knowledge Injection, which incorporates well-established placement knowledge, and Expert Workflow Imitation, which emulates the post-refinement process of human experts. The effectiveness of EXPlace is demonstrated on the ICCAD 2015 and OpenROAD benchmarks.

**Strengths:**

The paper is well-written, clear, and easy to follow;

The guidance from expert knowledge is effective, the results are impressive, and the selection of expert knowledge is reasonable.

**Weaknesses:**

The main contribution of this paper lies in incorporating expert knowledge into the training process of reinforcement learning. It is well known that introducing expert knowledge during training can significantly benefit the early stages of learning, enabling the model to quickly reach a relatively high level of performance. This has been demonstrated in many works, such as AlphaGo. Therefore, the conclusions of this paper are not particularly novel. On the other hand, although the paper achieves impressive results, this may be due to the fact that using reinforcement learning to solve macro placement is still at an early stage. In such cases, the involvement of expert knowledge can lead to a noticeable improvement. However, one of the goals of AI is to surpass the limitations of human expert knowledge in order to achieve better performance. For example, algorithms like AlphaZero entirely abandoned expert priors and reached even higher levels of performance. In such scenarios, expert knowledge can become a local optimum that restricts further breakthroughs. Thus, I believe this paper represents a solid engineering effort within the current technological context, but its conclusions and level of innovation are limited in the long term. For other issues, please refer to the Question section.

**Questions:**

1. The expert knowledge is introduced as a cost signal in reinforcement learning and combined through linear weighting. Are there any conflicts between these different pieces of expert knowledge? Moreover, if more expert knowledge is incorporated, is this cost-summing approach still feasible?

2. How does the training cost change?

3. Can you provide a few examples of layout comparisons to demonstrate that the resulting layouts indeed reflect the intended effects of the expert constraints, rather than just improvements in evaluation metrics?

---

> ### Author Response · Authors · 2025-11-21
>
> Thanks for dedicating your time to reviewing our paper! Your suggestions are very constructive for us to further improve the paper. According to your suggestions, we have revised our paper and uploaded the updated version. All modifications are highlighted in red for your convenience. Please find our detailed responses to your comments below.
>
> ## Response to weakness "It is well known that introducing expert knowledge during training can significantly benefit the early stages of learning,......, the conclusions of this paper are not particularly novel."
>
> Thank you for your insightful and constructive feedback. We would like to clarify that the conceptual contribution of this work is indeed novel and significant within the AI for electronic design automation (EDA) domain. Specifically, our paper is the first to successfully integrate expert knowledge from EDA methods with advanced deep reinforcement learning (RL) techniques from the AI domain to address the challenging macro placement problem (which is more complex than the the well-structured *Go* game, as elaborated in the next response), and achieves impressive power, performance, and area (PPA) improvements over both AI-based and traditional EDA approaches.
>
> The early work of RL for macro placement is AlphaChip [1], a purely data-driven method published in *Nature 2021*, which encodes the macro netlist using graph neural networks and employs delayed rewards based on half-perimeter wirelength (HPWL) and congestion. However, due to the indistinguishable feature encoding and overly sparse rewards, its performance is often suboptimal and can be easily surpassed by later black-box optimization (BBO) methods [2]. Subsequent works, such as Cheng et al. [3] and Lai et al. [4], introduced vision-based encodings of chip layouts and extended the input space by incorporating wiremask techniques, thus improving the correlation between features and the step-wise HPWL increments. More recently, Xue et al. [5] proposed to refine the solutions produced by traditional methods and enhance their regularity to reduce routing congestion. These developments gradually enabled RL-based methods to become competitive with modern BBO placers [7] and mixed-size analytical placers [6].
>
> However, most of these approaches still operate under oversimplified objective formulations, such as minimizing HPWL that correlates poorly with final PPA metrics, and allow the RL agent to generate placement solutions freely without considering high-level constraints. This hinders the progress of data-driven AI methods. Under such a misaligned formulation, no matter how powerful the AI (RL) algorithm is, it is doomed to fail in achieving industrial-grade results, as the system lacks the necessary guidance to distinguish between truly effective and suboptimal placements, and the input features provide insufficient support for such differentiation.
>
> Therefore, at the current stage of RL-based chip placement research, we believe the incorporation of expert knowledge to enhance both the learning framework and its objective formulation represents a fundamental and necessary step forward. Our work takes an early step in this direction and demonstrates substantial empirical gains, thereby validating its effectiveness and correctness. We believe this conceptual contribution is both timely and impactful, offering a promising direction that may inspire future research in the field.
>
> Once again, we appreciate your thoughtful comments. We hope our response can address your concerns.

---

> ### Author Response · Authors · 2025-11-21
>
> ## Response to weakness "expert knowledge can become a local optimum that restricts further breakthroughs......its conclusions and level of innovation are limited in the long term."
> Thank you for the insightful comments. For the macro placement problem, we do not believe that AI alone is currently capable of producing industrial-grade results or even surpassing experienced human engineers. In practice, most leading chip design houses still rely on human expertise to determine macro locations, which remains poorly automated and can take days or even weeks to finalize. At the current stage of research, it is more practical and reliable to emulate expert strategies and reduce manual effort in macro placement than to eliminate human involvement entirely.
>
> However, we fully agree that the long-term goal of AI is to go beyond human expert knowledge and achieve even better performance. This has been demonstrated in well-structured domains such as the game of *Go*, where systems like AlphaZero have outperformed top human players without using expert knowledge or imitating human decisions. In this context, we would like to point out that the chip placement problem is fundamentally different from games like Go, which is summarized in the following comparison table.
>
> | Features | Scale of Search Space | Expensive or Black-Box Objective | Dependencies with Upstream and Downstream Tasks |
> | --- | --- | --- | --- |
> | Go | $10^{360}$ | x | x |
> | Macro Placement | $>10^{2500}$ [1] | ✓ | ✓ |
>
> As shown above, *Go* has a clear and complete problem setting with well-defined rules and objectives. In contrast, macro placement involves a larger search space, expensive PPA evaluations, and dependencies on upstream and downstream design stages. Placement is only one local stage in the full physical design flow of integrated circuits, which also includes logic synthesis, floorplanning, timing optimization, clock tree synthesis, routing, and sign-off. These stages are closely nested, and optimizing one stage in isolation usually does not lead to satisfactory results in terms of PPA. Moreover, the stages following placement are time-consuming and computationally expensive, so getting direct PPA feedback is expensive and inefficient during training.
>
> The success of AlphaZero benefits from efficient self-play and a reliable value network that guides exploration of the search space. However, when solving macro placement, it is extremely difficult to construct a value network that accurately models the impact of each action on final PPA, due to the absence of information from other design stages and the high cost of obtaining ground-truth PPA feedback. Under such a “blind” setting, even a system as powerful as AlphaZero may struggle to perform effectively.
>
> Therefore, integrating expert knowledge into AI-based placers is not just beneficial for the early stage of learning but essential for industrial deployment. It provides critical inductive biases and domain knowledge that can guide the learning process, reduce misalignment, and improve both efficiency and solution quality in this highly specialized task. In our work, we take this step for the first time by integrating expert knowledge from EDA methods into the RL framework. These expert priors, grounded in long-standing industry experience, are aware of the trade-offs involved throughout the full design flow. We believe this is a practical and necessary step toward making RL more relevant and effective in real-world chip design. In addition, we explored preference-based optimization techniques, which allow us to consider downstream metrics more directly and with better sample efficiency.
>
> Although our current results may not yet surpass manually optimized placements, they do outperform traditional expert-designed heuristics (e.g., Hier-RTLMP [7]), which shows another way RL with expert guidance can exceed human baselines. Looking ahead, we believe that it is possible to totally surpass manual placements through two technical roadmaps: (1) training accurate surrogate models for PPA metrics, which would enable direct optimization, though data scarcity remains a major challenge due to the limited open-source chip designs; and (2) further enhancing expert-guided AI systems, in which AI can leverage its high search efficiency and cross-design generalization ability to find better layouts or achieve expert-level results more efficiently.
>
> Thank you again for your valuable comments, which have helped us better explain the motivation and contributions of our work. We hope this response can address your concerns.

---

> ### Author Response · Authors · 2025-11-21
>
> ## Response to "... Are there any conflicts between these different pieces of expert knowledge? if more expert knowledge is incorporated, is this cost-summing approach still feasible? "
>
> Yes, conflicts are inherent when multiple expert priors are introduced as cost signals. For instance, the wiremask and dataflow mask encourage macro clustering to reduce the length of interconnectation, while the periphery mask disperses macros toward the boundary; the I/O keepout mask can further contradict periphery preference by banning specific edge regions. This tension is not unique to our method, which is a fundamental issue for any approach that attempts broad coverage of expert priors.
>
> Despite these conflicts, the linear weighted-sum formulation remains the prevailing and practical choice. Classical macro placers adopt this approach explicitly (e.g., Eq. (3) in [7], Eq. (8) in [8], Eq. (6) in [9]), and both [7] and [9] successfully incorporate a broader set of expert knowledge while continuing to rely on linear aggregation. In practice, prior work provides guidance on how to prioritize specific expert objectives [7]. Beyond such well-informed default configurations, Bayesian optimization, as employed in [9], offers a systematic means of exploring the weight space.
>
> In our experiments, we follow the guidance provided in [7] to set the trade-off weights, which has already yielded substantial performance improvements without requiring extensive manual tuning. We acknowledge that incorporating additional expert priors increases the overhead of hyperparameter tuning and, in some edge cases, conflicting signals may degrade overall performance. As immediate solutions, we can either employ black-box hyperparameter tuning to adapt the weights for each design individually, or leverage advanced multi-task learning techniques to mitigate conflicts among objectives.
>
> Thank you for your thoughtful question. We hope this response can address your concerns.
>
>
> ## Response to "How does the training cost change?"
> Thank you for your thoughtful question. In response, we have revised our paper to provide the reward and cost function curves in Figure 10 of Appendix D.7. It can be observed that the total scaled reward converges in a stable manner, and the costs associated with HPWL, dataflow, macro grouping, and peripheral biasing all decrease, demonstrating the effectiveness of our reinforcement learning training. The only exception is the I/O keepout cost, which exhibits fluctuations during training. This behavior may result from the relatively low weight assigned to the I/O keepout term to encourage flexible placement, as well as its intrinsic conflict with peripheral biasing—given that I/O ports are distributed along the periphery. Consequently, improvements in peripheral biasing may negatively impact the I/O keepout cost. Despite the fluctuations in I/O keepout cost, the bottom visualization of Figure 10 indicates that the policy does learn to avoid I/O ports by placing more macros on the top side of the layout, rather than the bottom, which is densely populated with I/O ports.
>
> Thank you again for your question. We hope this response can provide more information and address your concerns.

---

> ### Author Response · Authors · 2025-11-21
>
> ## Response to "...provide a few examples of layout comparisons..."
>
> Thank you for your insightful question. We would like to draw your attention to the placement visualizations provided in Figures 8 and 9 of Appendix D.6, which include two representative cases. In these figures, macros sharing the same color represent members of the same group, while the red-shaded regions denote the I/O keepout zones.
>
> In comparison to the layout generated by MaskRegulate [5], our proposed method EXPlace demonstrates a more effective capture of grouping intent. Specifically, EXPlace not only preserves the peripheral placement preference but also ensures that macros belonging to the same group are placed in close proximity. In contrast, MaskRegulate primarily emphasizes peripheral placement, resulting in the dispersion of grouped macros across different regions. When compared with LaMPlace [10] in Figure 9, EXPlace exhibits a notable advantage by reserving a large contiguous whitespace near the I/O ports. The whitespace is well utilized by placed cells, and thus, reduces routing detours, as evidenced by the reduced number of inserted buffers (colored in red points).
>
> The visualization of dataflow optimization remains challenging due to the high density of connections. It is difficult to discern qualitative differences from visualizations alone. Instead, we provided numerical evidence of optimizing dataflow cost in Figure 10 of Appendix D.7. Specifically, the training cost curve shows a clear decreasing trend in dataflow cost throughout the training process, demonstrating the effectiveness of EXPlace in optimizing dataflow.
>
> We hope this response can address your concerns, and we sincerely appreciate your thoughtful engagement with our work.
>
>
> ***Refs***
>
> [1] A graph placement methodology for fast chip design. Nature, 2021.
>
> [2] Macro placement by wire-mask-guided black-box optimization. NeurIPS, 2023.
>
> [3] On joint learning for solving placement and routing in chip design. NeurIPS, 2021.
>
> [4] MaskPlace: Fast chip placement via reinforced visual representation learning. NeurIPS, 2022.
>
> [5] Reinforcement learning policy as macro regulator rather than macro placer. NeurIPS, 2024.
>
> [6] DREAMPlace 4.0: Timing-driven placement with momentum-based net weighting and Lagrangian-based refinement. IEEE TCAD, 2023.
>
> [7] Hier-RTLMP: A hierarchical automatic macro placer for large-scale complex IP blocks. IEEE TCAD, 2023.
>
> [8] Regularity-Aware Routability-Driven Macro Placement Methodology for Mixed-Size Circuits With Obstacles. IEEE TVLSI 2019.
>
> [9] RTL-MP: Toward Practical, Human-Quality Chip Planning and Macro Placement. ISPD 2022.
>
> [10] LaMPlace: Learning to optimize cross-stage metrics in macro placement. ICLR, 2025.

---

### Official Review · Reviewer_ocTp · 2025-10-30

**Soundness:** 3
**Presentation:** 3
**Contribution:** 3
**Rating:** 6
**Confidence:** 3

**Summary:**

This paper presents an RL-based macro placement framework that integrates expert knowledge and workflow imitation, achieving significant PPA improvements over existing methods. The approach is innovative and practical, effectively bridging the gap between data-driven optimization and human design expertise.

**Strengths:**

- The experiments are very extensive across different placement benchmarks.

- The results are promising, it can achievethe  best placement results in different cases.

**Weaknesses:**

- The main contribution is to add three more expert masks, while adding different masks is not a very new idea. It has been attempted in previous work as MaskRegulator.

- According to Fig. 6 (a), the computation of the Periphery mask and the Dataflow mask is very time-consuming, resulting in relatively low sample efficiency.

**Questions:**

- How to collect the preference pairs D in the timing preference fine-tuning in ICCAD2015 or OpenROAD?
- Are there any ablation results with and without timing-driven fine-tuning?
- Why are there no DRC results in the ICCAD2015 benchmark?

---

> ### Author Response · Authors · 2025-11-21
>
> Thanks for dedicating your time to reviewing our paper! Your suggestions are very constructive for us to further improve the paper. According to your suggestions, we have revised our paper and uploaded the updated version. All modifications are highlighted in red for your convenience. Please find our detailed responses to your comments below.
>
> ## Response to "The main contribution is to add three more expert masks, while adding different masks is not a very new idea."
>
> Thank you for your insightful comment. We agree that adding masks is not a very new idea. But we would like to clarify that the primary contribution of our work does not lie merely in the addition of expert masks from a technical novelty perspective. Rather, our focus is on a ***conceptual advancement***: the integration of human expert knowledge into the data-driven framework for industrial-level chip placement. Since the publication of AlphaChip [1] in *Nature*, many follow-up works [2,3,4] have sought to improve RL-based placement methods through better representation learning or the design of dense reward signals. However, most of these approaches still operate under oversimplified objective formulations, such as minimizing HPWL, and allow the RL agent to generate placement solutions without considering high-level constraints. Under such a misaligned formulation, no matter how powerful the AI (RL) algorithm is, it is doomed to fail in achieving industrial-grade results. Therefore, at this stage of RL-based placement, we believe the incorporation of expert knowledge to enhance both the framework and its objective formulation represents a fundamental and necessary step forward. Our work takes an early step in this direction and demonstrates substantial empirical gains, thereby validating the effectiveness of integrating expert knowledge. We believe this conceptual contribution is both timely and impactful, offering a promising direction that may inspire future research in the field.
>
> From a technical standpoint, identifying relevant expert knowledge and translating it into step-wise masks and decomposable rewards is non-trivial. For example, in the Hier-RTL-MP method [5], the utilized expert knowledge was originally embedded in black-box evaluations or rules that only assess final placement outcomes and provide no intermediate feedback. Our technical contribution lies in the design of appropriate representations and decomposable reward functions, which allowed us to successfully integrate these expert signals into the RL framework. Furthermore, we also explored preference-based optimization strategies to refine cross-stage objectives with improved sample efficiency, i.e., requiring fewer evaluations of costly objectives. This addition further enhances the practicality and scalability of our approach.
>
> In summary, while the idea of using multiple masks is not entirely new, our work advances the field through a novel conceptual contribution and practical realization of expert knowledge integration within RL-based placement. We hope this can address your concerns and appreciate the opportunity to clarify this point. Thank you again for the constructive feedback.
>
> ## Response to "the Periphery mask and the Dataflow mask is very time-consuming"
>
> Thank you for your insightful and constructive comment. We fully acknowledge that the computation of the Periphery mask and Dataflow mask introduces additional overhead. However, we believe that this overhead is not that significant since: (1) The mask computation can be refined by parallelization or GPU acceleration. (2) Under generalization settings, the additional runtime introduced by expert masks is orders of magnitude smaller than the runtimes of black-box optimization (BBO)-based and analytical placement methods. Hence, this overhead is relatively minor, especially when weighed against the performance gains enabled by the expert masks.
>
> (Please see the next comment)

---

> ### Author Response · Authors · 2025-11-21
>
> (Following up on the previous comment)
>
> As discussed in Appendix D.2, we have mitigated the runtime issue by parallelizing the environment transition process (including the mask computation) across multiple episodes. To further clarify the runtime efficiency, we have updated Appendix D.2 with more comprehensive runtime analyses. The revised results (also shown in the table below for your convenience) demonstrate that the overall training time of EXPlace is, in fact, faster than that of MaskRegulate [4]. Compared to MaskPlace [3], although EXPlace still incurs a higher runtime, we believe this is a reasonable trade-off considering the substantial performance improvements reported in Tables 1 and 2.
>
> Table R1. Runtime (in seconds) of various methods on the ICCAD 2015 benchmark. The runtime of LaMPlace is reported as in the original paper.
>
> | Method             | Time Type | superblue1 | superblue3 | superblue4 | superblue5 | superblue7 | superblue10 | superblue16 | superblue18 | Avg.     |
> |--------------------|-----------|------------|------------|------------|------------|------------|-------------|-------------|-------------|----------|
> | DREAMPlace         | Test      | 287.30     | 158.11     | 107.54     | 121.20     | 199.03     | 210.90      | 112.70      | 66.12       | 157.86   |
> | LaMPlace           | Test      | 792.00     | 1368.00    | 396.00     | 2592.00    | 756.00     | 11376.00    | 36.00       | 180.00      | 2187.00  |
> | MaskPlace          | Train     | 3376.95    | 4755.24    | 6369.17    | 4514.60    | 11975.79   | 4949.54     | 2623.18     | 2571.54     | 5142.00  |
> |                    | Test      | 1.64       | 2.70       | 3.88       | 2.28       | 5.93       | 3.46        | 2.59        | 3.11        | 3.20     |
> | MaskRegulate       | Train     | 8534.70    | 12636.81   | 15743.53   | 11497.57   | 30039.77   | 10109.45    | 6122.82     | 6381.81     | 12633.31 |
> |                    | Test      | 4.69       | 8.08       | 8.92       | 6.89       | 17.95      | 8.09        | 3.37        | 3.72        | 7.72     |
> | EXPlace (Ours)     | Train     | 6984.49    | 8384.76    | 11036.91   | 7723.30    | 15939.91   | 8696.94     | 5178.73     | 5051.55     | 8624.57  |
> |                    | Test      | 5.28       | 9.31       | 10.68      | 6.80       | 19.55      | 7.49        | 5.03        | 4.97        | 8.64     |
>
>
> Moreover, in comparison to BBO-based and analytical approaches, RL methods like EXPlace exhibit substantially faster inference times. For instance, on the superblue1 benchmark, EXPlace achieves an inference time of 5.28 seconds, whereas the evolutionary optimization of LaMPlace [6] requires 792 seconds. This large disparity highlights the practical efficiency of RL-based methods during the deployment phase. Consequently, under generalization settings, the additional runtime associated with expert mask computation becomes less significant.
>
> (Please see the next comment)

---

> ### Author Response · Authors · 2025-11-21
>
> (Following up on the previous comment)
>
> Additionally, we would like to draw attention to the new results presented in Table 7 of the revised paper (also reproduced in the table below for reference), which compare time and performance under generalization settings. These results show that, with much less runtime, zero-shot EXPlace (only trained on superblue1) consistently outperforms BBO-based and analytical methods in both HPWL and WNS metrics, a level of generalization performance that MaskPlace [3] and MaskRegulate [4] fail to achieve. We believe these findings further support the trade-off between mask computation time and overall performance gains.
>
> Table R2. Comparison of generalization performance on the ICCAD 2015 benchmarks. RL models trained on superblue1 are evaluated on other cases in a zero-shot manner. BBO and analytical methods are also included for comparison. Runtimes are reported in seconds.
>
> | Method         | Metric  | superblue1 | superblue3 | superblue4 | superblue5 | superblue7 | superblue10 | superblue16 | superblue18 | Avg. Rank |
> |----------------|---------|------------|------------|------------|------------|------------|-------------|-------------|-------------|------------|
> | DREAMPlace     | HPWL    | 768.5      | 971.3      | 673.7      | 824.3      | 1063.3     | 1200.9      | 665.1       | **256.8**   | 4.50       |
> |                | WNS     | -67.50     | -106.86    | -67.35     | **-57.50** | -101.27    | -264.08     | -36.57      | -22.12      | 3.38       |
> |                | TNS     | -1389.24   | -1187.80   | -1044.56   | **-673.01**| **-520.82**| **-2583.61**| **-1011.32**| **-52.48**  | **1.50**   |
> |                | Runtime | 287.30     | 158.11     | 107.54     | 121.20     | 199.03     | 210.90      | 112.70      | 66.12       | 4.12       |
> | LaMPlace       | HPWL    | 577.7      | 845.5      | 467.3      | **645.3**  | 924.9      | 1050.5      | 498.7       | 281.6       | 2.62       |
> |                | WNS     | -56.37     | -174.94    | -44.75     | -158.91    | -88.09     | **-77.54**  | **-34.68**  | -47.41      | 2.88       |
> |                | TNS     | -1536.77   | -1797.70   | -1053.89   | -1300.71   | -1602.31   | -3441.67    | -1527.76    | -311.15     | 3.38       |
> |                | Runtime | 792.00     | 1368.00    | 396.00     | 2592.00    | 756.00     | 11376.00    | 36.00       | 180.00      | 4.88       |
> | MaskPlace      | HPWL    | 733.8      | 822.8      | 505.7      | 784.5      | 981.4      | 1006.5      | 531.6       | 328.2       | 3.75       |
> |                | WNS     | -77.13     | -98.61     | -48.88     | -192.60    | **-77.44** | -108.36     | -42.04      | -31.49      | 3.38       |
> |                | TNS     | -1644.56   | -1475.26   | -1452.97   | -2003.23   | -1771.30   | -4191.77    | -1763.34    | -219.48     | 4.38       |
> |                | Runtime | 1.64       | 2.70       | 3.88       | 2.28       | 5.93       | 3.46        | 2.59        | 3.11        | **1.00**   |
> | MaskRegulate   | HPWL    | 600.9      | 761.2      | 432.4      | 710.1      | 910.0      | 1190.9      | 517.0       | 291.6       | 2.88       |
> |                | WNS     | -78.50     | **-86.66** | -53.46     | -148.92    | -93.97     | -77.59      | -39.06      | -31.34      | 3.12       |
> |                | TNS     | -1093.88   | -1330.60   | -1253.22   | -1414.67   | -1826.75   | -4904.24    | -1421.76    | -217.42     | 3.38       |
> |                | Runtime | 4.69       | 8.08       | 8.92       | 6.89       | 17.95      | 8.09        | 3.37        | 3.72        | 2.25       |
> | EXPlace (Ours) | HPWL    | **536.9**  | **698.9**  | **393.1**  | 666.2      | **871.8**  | **862.6**   | **472.8**   | **270.1**   | **1.25**   |
> |                | WNS     | **-51.25** | -101.04    | **-39.76** | -140.89    | -87.10     | -99.97      | -43.40      | **-21.77**  | **2.25**   |
> |                | TNS     | **-1070.72**| **-1063.76**| **-820.35**| -1462.69 | -1479.43   | -3444.93    | -1999.02    | -140.40     | 2.38       |
> |                | Runtime | 5.28       | 9.31       | 10.68      | 6.80       | 19.55      | 7.49        | 5.03        | 4.97        | 2.75       |
>
>
> We hope these additional analyses can address your concerns and sincerely thank you for pointing out this important aspect.

---

> ### Author Response · Authors · 2025-11-21
>
> ## Response to "How to collect the preference pairs D?"
> Thank you for your insightful question. We apologize for the omission of details regarding the construction of preference pairs in the original paper. In response to your question, we have revised Section 4.2 to provide a clearer explanation of the preference pair collection process.
>
> Specifically, our preference optimization is designed as a self-supervised process. During each fine-tuning iteration, multiple trajectories are sampled from the EXPlace policy. These trajectories are then evaluated using OpenTimer to obtain their timing performance. Based on these evaluations, the trajectories are partitioned into preferred and rejected sets, forming the preference pairs *D*. To mitigate the effect of noise in the ranking, we retain only the top 10% of preference pairs that exhibit sufficiently distinguishable timing differences, as described in Appendix C.
>
> We sincerely appreciate your question and hope that these clarifications address your concerns.
>
> ## Response to "Are there any ablation results with and without timing-driven fine-tuning?"
>
> Thank you for your insightful question. We apologize for the lack of clarity and completeness regarding this part of the experiments in the original version of our paper. In Figure 5, we presented the results with and without timing-driven fine-tuning, denoted as "Full iterations" and "Initial policy", respectively, on three representative cases, which demonstrate the effectiveness of DPO in improving timing performance. In addition, to address your concerns more thoroughly, we have revised the paper to include a more comprehensive comparison of EXPlace with and without timing-driven fine-tuning, as shown in Table 9 of Appendix D.5 (also reproduced in the table below for your convenience). The updated results indicate that the timing-driven fine-tuning process leads to significant improvements in both WNS and TNS, by 22.68% and 12.29%, respectively. Moreover, it can be observed that the overall reduction in global HPWL is marginal. This contrast suggests that the fine-tuning process primarily targets the optimization of critical timing paths to improve TNS and WNS, rather than simply minimizing HPWL.
>
> Table R3: Comparison of EXPlace with and without timing-driven fine-tuning (i.e., DPO) on ICCAD 2015 contest benchmarks. The evaluation metrics include Global HPWL (m), TNS (×10² ns), and WNS (ns). The best results are highlighted in **bold**.
>
> | Design        | TNS (EXPlace) | WNS (EXPlace) | HPWL (EXPlace) | TNS (EXPlace + DPO)  | WNS (EXPlace + DPO)  | HPWL (EXPlace + DPO) |
> |---------------|---------------|----------------|------------------|-------------|-------------|--------------|
> | superblue1    | -1070.14      | -51.48         | **534.00**       | **-821.20** | **-36.48**  | 559.23       |
> | superblue3    | -734.65       | -86.83         | 654.10           | **-682.85** | **-74.16**  | **637.13**   |
> | superblue4    | -668.98       | -39.48         | 382.10           | **-474.97** | **-22.55**  | **377.35**   |
> | superblue5    | -1060.71      | -138.45        | 690.30           | **-867.83** | **-98.52**  | **650.10**   |
> | superblue7    | -1227.42      | -61.34         | 832.90           | **-1045.91**| **-47.33**  | **810.48**   |
> | superblue10   | **-2168.76**  | -96.34         | 969.70           | -2499.09    | **-65.08**  | **892.20**   |
> | superblue16   | -339.35       | -17.62         | **449.10**       | **-284.59** | **-17.09**  | 449.97       |
> | superblue18   | -106.30       | -21.69         | **269.90**       | **-100.81** | **-20.01**  | 273.24       |
> | **Avg. Improvement** | --           | --              | --               | **12.29%**  | **22.68%**  | **1.77%**    |
>
> Thank you again for your valuable question. We hope these new results help clarify the effectiveness of our approach.

---

> ### Author Response · Authors · 2025-11-21
>
> ## Response to "Why are there no DRC results in the ICCAD2015 benchmark?"
> Thank you for your thoughtful question. We sincerely apologize for the lack of clarity regarding this experimental detail in our original submission. To clarify, OpenTimer provides a ***post-placement*** timing evaluation, rather than a post-routing one, whereas design rule check (DRC) violations arise after the detailed routing process. We acknowledge that the absence of post-routing results on the ICCAD2015 benchmark affects the overall comprehensiveness of our experimental evaluation. In response, we have revised the paper to include updated post-routing results on ICCAD2015, which are now presented in Table 10 and Table 11. Specifically, Table 10 reports the results under the case-by-case training setting, while Table 11 presents the generalization performance. The post-routing performance is evaluated using the commercial tool *Cadence Innovus*, where the metrics of horizontal and vertical routing overflow (rOverflowH and rOverflowV) are analogous to DRC violations.
>
> Under the case-by-case training setting (see Table 10 of Appendix D.8), our proposed EXPlace achieves state-of-the-art results in rWL, NVP, WNS, and TNS, indicating its strong capability in optimizing both wirelength and timing metrics. While the rOverflowV of EXPlace is slightly inferior to that of MaskRegulate [4], its superior overall performance in NVP and timing reflects the high quality and spatial regularity of its placement solutions. Notably, compared to the runner-up method (i.e., LaMPlace [6]) in terms of TNS, EXPlace achieves an average improvement of 31.3%, which can substantially ease timing closure in practical design.
>
> Under the challenging generalization scenario (as presented in Table 11 of Appendix D.8), our method, EXPlace, trained solely on superblue1, consistently achieves superior average performance across key metrics, including rWL, NVP, and TNS, surpassing all RL-based, BBO-based, and analytical baselines. In particular, when compared to the analytical approach DREAMPlace [7], EXPlace demonstrates significantly reduced routing overflow, attributable to its enhanced placement regularity. This improvement in placement quality subsequently leads to better routed wirelength and timing performance.
>
> We hope these new results can address your concerns, and thank you very much again for your thoughtful questions.
>
> ***Refs***
>
> [1] A graph placement methodology for fast chip design. Nature, 2021.
>
> [2] On joint learning for solving placement and routing in chip design. NeurIPS, 2021.
>
> [3] MaskPlace: Fast chip placement via reinforced visual representation learning. NeurIPS, 2022.
>
> [4] Reinforcement learning policy as macro regulator rather than macro placer. NeurIPS, 2024.
>
> [5] Hier-RTLMP: A hierarchical automatic macro placer for large-scale complex IP blocks. IEEE TCAD, 2023.
>
> [6] LaMPlace: Learning to optimize cross-stage metrics in macro placement. ICLR, 2025.
>
> [7] DREAMPlace 4.0: Timing-driven placement with momentum-based net weighting and Lagrangian-based refinement. IEEE TCAD, 2023.

---

### Official Review · Reviewer_yPJt · 2025-10-31

**Soundness:** 3
**Presentation:** 3
**Contribution:** 2
**Rating:** 4
**Confidence:** 4

**Summary:**

This paper presents EXPlace, a reinforcement learning method for chip macro placement that systematically integrates domain expertise from EDA. The authors identify a key limitation in prior RL-based methods: their reliance on oversimplified proxy objectives, which leads to suboptimal results compared to human expert designs. EXPlace introduces Expert Knowledge Injection and Expert Workflow Imitation to address this problem. The method is evaluated on ICCAD 2015, demonstrating better performance compared to baseline methods.

**Strengths:**

1. This paper leverages the prior knowledge of human experts to enhance the final chip layout outcomes in this specialized domain.
2. Well-written and easy to read.

**Weaknesses:**

1. The runtime analysis is narrow, comparing EXPlace primarily against other RL methods. A more convincing efficiency demonstration requires benchmarking against a wider range of modern placers, including advanced analytical and black-box optimization methods.
2. For a comprehensive sign-off quality assessment, it is critical to include key industrial metrics like post-route power consumptionand final core area utilization.
3. The ablation study effectively tests several expert masks but omits a critical component: the periphery biasing mask.
4. The generalization test—training on one circuit and testing on four others from the same ICCAD 2015 benchmark—is promising but insufficient. Performance on the remaining three circuitsis unknown.

**Questions:**

See in weakness.

---

> ### Author Response · Authors · 2025-11-21
>
> Thanks for dedicating your time to reviewing our paper! Your suggestions are very constructive for us to further improve the paper. According to your suggestions, we have revised our paper and uploaded the updated version. All modifications are highlighted in red for your convenience. Please find our detailed responses to your comments below.
>
> ## Response to "runtime analysis is narrow, ..."
> Thank you for your insightful and constructive comment. In response, we have revised the paper to include a more comprehensive runtime analysis, as detailed in Appendix D.2. As you suggested, we expanded our comparison beyond RL-based methods to include a broader set of modern placers, including both advanced analytical and black-box optimization (BBO) approaches. The results are also shown in the table below for your convenience.
>
> Table R1. Runtime (in seconds) of various methods on the ICCAD 2015 benchmark. The runtime of LaMPlace is reported as in the original paper.
>
> | Method             | Time Type | superblue1 | superblue3 | superblue4 | superblue5 | superblue7 | superblue10 | superblue16 | superblue18 | Avg.     |
> |--------------------|-----------|------------|------------|------------|------------|------------|-------------|-------------|-------------|----------|
> | DREAMPlace         | Test      | 287.30     | 158.11     | 107.54     | 121.20     | 199.03     | 210.90      | 112.70      | 66.12       | 157.86   |
> | LaMPlace           | Test      | 792.00     | 1368.00    | 396.00     | 2592.00    | 756.00     | 11376.00    | 36.00       | 180.00      | 2187.00  |
> | MaskPlace          | Train     | 3376.95    | 4755.24    | 6369.17    | 4514.60    | 11975.79   | 4949.54     | 2623.18     | 2571.54     | 5142.00  |
> |                    | Test      | 1.64       | 2.70       | 3.88       | 2.28       | 5.93       | 3.46        | 2.59        | 3.11        | 3.20     |
> | MaskRegulate       | Train     | 8534.70    | 12636.81   | 15743.53   | 11497.57   | 30039.77   | 10109.45    | 6122.82     | 6381.81     | 12633.31 |
> |                    | Test      | 4.69       | 8.08       | 8.92       | 6.89       | 17.95      | 8.09        | 3.37        | 3.72        | 7.72     |
> | EXPlace (Ours)     | Train     | 6984.49    | 8384.76    | 11036.91   | 7723.30    | 15939.91   | 8696.94     | 5178.73     | 5051.55     | 8624.57  |
> |                    | Test      | 5.28       | 9.31       | 10.68      | 6.80       | 19.55      | 7.49        | 5.03        | 4.97        | 8.64     |
>
>
> Our updated analysis highlights an important distinction in evaluating runtime efficiency: RL-based methods, such as EXPlace, typically involve a substantial training cost but offer extremely fast inference during deployment. Therefore, we propose evaluating the trade-off between runtime and performance from two complementary perspectives:
>
> 1. Case-by-case optimization: Similar to BBO and analytical methods, RL algorithms can be trained from scratch for each individual case.
>
> 2. Generalization (zero-shot inference): A pre-trained RL policy is directly applied to unseen placement cases without retraining.
>
> In the original Table 1, we focused on the first perspective and showed that although EXPlace requires more time for training, it significantly outperforms BBO and analytical methods in terms of placement quality. This trade-off is often acceptable in industrial scenarios, where design cycles typically span weeks or even months, and higher placement quality justifies the additional training time.
>
> (Please see the next comment)

---

> ### Author Response · Authors · 2025-11-21
>
> (Following up on the previous comment)
>
> To address the second perspective, we have added new experiments in Appendix D.3, where we evaluate the zero-shot performance of EXPlace trained on superblue1 across eight different cases. The results (also presented in the table below) show that the zero-shot EXPlace still outperforms both BBO and analytical methods in terms of HPWL and WNS, even without any case-specific retraining. Moreover, the inference time of EXPlace is notably efficient, e.g., 5.28 seconds on superblue1, which is orders of magnitude faster than the 287s of the analytical DREAMPlace [1] and the 792s of the BBO-based LaMPlace [2]. Although DREAMPlace achieves superior TNS, the overall good performance and significant computational efficiency of EXPlace demonstrate its superiority over existing BBO and analytical methods.
>
>
> Table R2. Comparison of generalization performance on the ICCAD 2015 benchmarks. RL models trained on superblue1 are evaluated on other cases in a zero-shot manner. BBO and analytical methods are also included for comparison. Runtimes are reported in seconds.
>
> | Method         | Metric  | superblue1 | superblue3 | superblue4 | superblue5 | superblue7 | superblue10 | superblue16 | superblue18 | Avg. Rank |
> |----------------|---------|------------|------------|------------|------------|------------|-------------|-------------|-------------|------------|
> | DREAMPlace     | HPWL    | 768.5      | 971.3      | 673.7      | 824.3      | 1063.3     | 1200.9      | 665.1       | **256.8**   | 4.50       |
> |                | WNS     | -67.50     | -106.86    | -67.35     | **-57.50** | -101.27    | -264.08     | -36.57      | -22.12      | 3.38       |
> |                | TNS     | -1389.24   | -1187.80   | -1044.56   | **-673.01**| **-520.82**| **-2583.61**| **-1011.32**| **-52.48**  | **1.50**   |
> |                | Runtime | 287.30     | 158.11     | 107.54     | 121.20     | 199.03     | 210.90      | 112.70      | 66.12       | 4.12       |
> | LaMPlace       | HPWL    | 577.7      | 845.5      | 467.3      | **645.3**  | 924.9      | 1050.5      | 498.7       | 281.6       | 2.62       |
> |                | WNS     | -56.37     | -174.94    | -44.75     | -158.91    | -88.09     | **-77.54**  | **-34.68**  | -47.41      | 2.88       |
> |                | TNS     | -1536.77   | -1797.70   | -1053.89   | -1300.71   | -1602.31   | -3441.67    | -1527.76    | -311.15     | 3.38       |
> |                | Runtime | 792.00     | 1368.00    | 396.00     | 2592.00    | 756.00     | 11376.00    | 36.00       | 180.00      | 4.88       |
> | MaskPlace      | HPWL    | 733.8      | 822.8      | 505.7      | 784.5      | 981.4      | 1006.5      | 531.6       | 328.2       | 3.75       |
> |                | WNS     | -77.13     | -98.61     | -48.88     | -192.60    | **-77.44** | -108.36     | -42.04      | -31.49      | 3.38       |
> |                | TNS     | -1644.56   | -1475.26   | -1452.97   | -2003.23   | -1771.30   | -4191.77    | -1763.34    | -219.48     | 4.38       |
> |                | Runtime | 1.64       | 2.70       | 3.88       | 2.28       | 5.93       | 3.46        | 2.59        | 3.11        | **1.00**   |
> | MaskRegulate   | HPWL    | 600.9      | 761.2      | 432.4      | 710.1      | 910.0      | 1190.9      | 517.0       | 291.6       | 2.88       |
> |                | WNS     | -78.50     | **-86.66** | -53.46     | -148.92    | -93.97     | -77.59      | -39.06      | -31.34      | 3.12       |
> |                | TNS     | -1093.88   | -1330.60   | -1253.22   | -1414.67   | -1826.75   | -4904.24    | -1421.76    | -217.42     | 3.38       |
> |                | Runtime | 4.69       | 8.08       | 8.92       | 6.89       | 17.95      | 8.09        | 3.37        | 3.72        | 2.25       |
> | EXPlace (Ours) | HPWL    | **536.9**  | **698.9**  | **393.1**  | 666.2      | **871.8**  | **862.6**   | **472.8**   | **270.1**   | **1.25**   |
> |                | WNS     | **-51.25** | -101.04    | **-39.76** | -140.89    | -87.10     | -99.97      | -43.40      | **-21.77**  | **2.25**   |
> |                | TNS     | **-1070.72**| **-1063.76**| **-820.35**| -1462.69 | -1479.43   | -3444.93    | -1999.02    | -140.40     | 2.38       |
> |                | Runtime | 5.28       | 9.31       | 10.68      | 6.80       | 19.55      | 7.49        | 5.03        | 4.97        | 2.75       |
>
> (Please see the next comment)

---

> ### Author Response · Authors · 2025-11-21
>
> (Following up on the previous comment)
>
> Regarding the comparison with other RL methods, we acknowledge that EXPlace incurs slightly higher inference time due to the additional computation introduced by expert masks. However, we believe this overhead is justified by the resulting performance gains. Furthermore, the environment transition process, including mask computation, can be parallelized across multiple episodes during training, which helps to mitigate the overall training time. As shown in Table 6 (or Table R1 in the response), EXPlace achieves faster training than the original implementation of MaskRegulate [3], demonstrating the efficiency of our design. Note that the training runtime in the revised version is shorter than that reported in the original paper, as we exclude the time spent on evaluating intermediate results using DREAMPlace and OpenTimer.
>
> We hope this extended analysis addresses your concern and provides a more convincing evaluation of EXPlace’s trade-off between runtime and performance.
>
>
> ## Response to "..., it is critical to include key industrial metrics like post-route power consumption and final core area utilization"
> Thank you for your valuable feedback. In response, we have revised the paper to include comparisons of post-route power and area metrics, as presented in the updated Table 2. The new results demonstrate that our method, EXPlace, achieves the lowest power consumption and the smallest cell area among all compared approaches. For your convenience, the updated power and area results are also presented in the table below.
>
> Table R3: Partial comparison results on OpenROAD benchmarks. We report Power (W), Cell Area (×10⁴μm²) and average ranking.
>
> | **Method**         | **Metric**   | **ariane133** | **ariane136** | **bp**     | **bp_be**  | **bp_fe**  | **swerv_wrapper** | **Avg. Rank** |
> |--------------------|--------------|---------------|---------------|------------|------------|------------|-------------------|---------------|
> | **DREAMPlace**     | Power        | 0.4537        | 0.5182        | **0.4954** | 0.2025     | 0.2141     | 0.2701            | 4.00          |
> |                    | Cell Area    | **38.55**     | 40.04         | **52.72**  | 12.27      | 7.50       | 23.08             | 2.67          |
> | **Hier-RTLMP**     | Power        | 0.4481        | 0.4991        | 0.4985     | 0.2055     | 0.2130     | 0.2670            | 3.33          |
> |                    | Cell Area    | 38.88         | 40.06         | 53.35      | 12.34      | 7.43       | 23.31             | 4.00          |
> | **MaskPlace**      | Power        | 0.4512        | 0.4976        | 0.4957     | 0.2032     | 0.2152     | 0.2732            | 3.83          |
> |                    | Cell Area    | 38.58         | **39.81**     | 53.84      | 12.24      | 7.41       | 23.46             | 3.33          |
> | **MaskRegulate**   | Power        | 0.4494        | 0.5049        | 0.5501     | 0.2079     | 0.2120     | 0.2747            | 5.00          |
> |                    | Cell Area    | 39.69         | 40.80         | 53.80      | 12.52      | 7.29       | 23.25             | 4.67          |
> | **LaMPlace**       | Power        | 0.4477        | **0.4969**    | 0.5211     | 0.2033     | 0.2095     | **0.2656**         | 2.50          |
> |                    | Cell Area    | 38.86         | 40.26         | 54.00      | 12.39      | 7.37       | 22.91             | 4.00          |
> | **EXPlace (Ours)** | Power        | **0.4472**    | 0.4996        | 0.5038     | **0.2012** | **0.2083** | 0.2681            | **2.33**      |
> |                    | Cell Area    | 39.15         | 40.07         | 52.92      | **12.21**  | **7.27**   | **22.85**         | **2.33**      |
>
> The observed reduction in cell area (i.e., number of cells) can be attributed to two key factors:
>
> 1. IO keepout constraints preserve sufficient space for placing IO-driven cells near the IO ports, thereby eliminating the need for additional buffers to drive distant cells.
>
> 2. Dataflow-aware optimization and macro grouping facilitate the co-placement of macros that correspond to related logic or lie along the same data path. This not only shortens the overall wirelength but also reduces the number of buffer cells required to repair critical timing paths.
>
> As a result of the reduced cell count, the overall power consumption is also lowered. Furthermore, as illustrated in Figures 8 and 9, EXPlace produces highly regular placements, with macros uniformly distributed along the periphery and well-aligned with each other. This regularity enables a more balanced clock tree synthesis, which in turn requires fewer clock buffers and shorter clock wires. As highlighted by prior work [4], the clock tree can account for 30%–70% of total power consumption.
>
> We hope these additional analyses effectively address your concerns, and we sincerely thank you again for your constructive suggestions.

---

> ### Author Response · Authors · 2025-11-21
>
> ## Response to "The ablation study ... omits a critical component: the periphery biasing mask"
> Thank you for your valuable comment. We agree that incorporating the ablation of the periphery biasing mask would enhance the comprehensiveness of our evaluation. In response to your suggestion, we have conducted the additional ablation experiment and included the results in Table 5, Appendix D.2 of the revised paper. The results indicate that excluding the periphery biasing mask leads to a degradation in both HPWL and timing performance. This outcome can be attributed to increased macro congestion in the central region of the chip, which in turn lengthens interconnect distances and exacerbates timing-critical paths. The results of ablating periphery biasing are also shown in the table below for your convenience.
>
> Table R4: Ablation study on four representative ICCAD 2015 cases. "w/o" denotes "without".
>
> | **Method**                 | **Metric** | **superblue1** | **superblue3** | **superblue16** | **superblue18** |
> |----------------------------|------------|----------------|----------------|------------------|------------------|
> | **EXPlace (w/o Periphery)**| HPWL       | 566.8          | 695.0          | 460.2           | 271.0           |
> |                            | WNS        | -64.93         | -79.54         | -28.92          | **-14.14**       |
> |                            | TNS        | -1335.10       | -730.21        | -1165.77        | -137.23         |
> | **EXPlace**                   | HPWL       | **534.0**      | **654.1**      | 449.1           | **269.9**       |
> |                            | WNS        | **-51.48**     | -86.83         | **-17.62**      | -21.69          |
> |                            | TNS        | -1070.14       | -734.65        | **-339.35**     | **-106.30**     |
>
>
> We believe this additional analysis can strengthen the empirical support for our method and sincerely thank you for pointing out this important aspect.
>
> ## Response to "The generalization test ... is promising but insufficient"
>
> Thank you for your insightful and constructive feedback. We fully agree that a more comprehensive evaluation involving all available benchmark circuits would provide a more complete and convincing assessment. We have revised the paper to include the full generalization results across all remaining circuits. These results are now reported in Table 7 of Appendix D.3 (or Table R2 in the response). As shown, our proposed method EXPlace consistently outperforms other RL baselines in terms of HPWL, WNS, and TNS, under the challenging setting of cross-design generalization. We attribute this performance to the robustness and transferability of the expertise-constrained decision employed by EXPlace. Notably, we observe an interesting phenomenon on superblue10, where the zero-shot EXPlace (trained on superblue1) achieves even better HPWL than the version trained directly on superblue10. This suggests that leveraging cross-design knowledge may help escape local optima, which is a direction that we believe is worth further investigation in future work. Additionally, Table 7 also presents a comparison of runtime and performance across various RL and BBO (or analytical) methods, as discussed in our first response. We hope these extended results better address your concern and demonstrate the generalization capability and practical value of our approach.
>
> ## New experimental results of post-routing performance on ICCAD 2015 benchmarks
>
> For a more comprehensive evaluation, we have added the post-routing performance evaluation on the ICCAD2015 benchmarks, conducted using the commercial tool *Cadence Innovus*. The results are presented in Table 10 and Table 11 of Appendix D.8, where Table 10 reports the results under the case-by-case training setting, and Table 11 summarizes the generalization performance.
>
> Under the case-by-case training setting, EXPlace achieves state-of-the-art performance in both wirelength and timing metrics, notably outperforming LaMPlace [2] on TNS by 31.3%. In the generalization scenario, EXPlace, trained only on superblue1, consistently outperforms all RL, BBO, and analytical baselines. We hope that these additional experimental results further demonstrate the effectiveness of our approach. Thank you very much again for your time and thoughtful review.
>
>
> ***Refs***
>
> [1] DREAMPlace 4.0: Timing-driven placement with momentum-based net weighting and Lagrangian-based refinement. IEEE TCAD, 2023.
>
> [2] LaMPlace: Learning to optimize cross-stage metrics in macro placement. ICLR, 2025.
>
> [3] Reinforcement learning policy as macro regulator rather than macro placer. NeurIPS, 2024.
>
> [4] Clock-Tree-Aware Incremental Timing-Driven Placement. ACM TODAES, 2016.

---

### Author Response · Authors · 2025-11-27

Dear Reviewers,

We sincerely thank you for your time and effort in reviewing our paper and for providing valuable and constructive feedback. In response to your insightful comments, we have made extensive efforts to include new experiments and extended discussions. The paper has also been revised accordingly, with the modified parts highlighted in red. We summarize our main updates and clarifications below.

As the discussion phase of ICLR is drawing to a close in one week, we sincerely look forward to your further feedback.

1. **Runtime Analysis (Reviewers yPJt and ocTp)**:
Following your suggestions, we conducted a more comprehensive runtime analysis that includes comparisons with various modern placers. Combined with the new generalization experiments, these results demonstrate that **EXPlace outperforms both BBO and analytical methods with much less runtime**.

2. **Novelty (Reviewer ocTp)**: We clarified the novelty and significance of integrating expert knowledge, emphasizing that it serves as a necessary and foundational step toward advancing RL-based macro placement approaches.

3. **AI vs. Human Expertise (Reviewer zEBD)**: We elaborated on the complexity of the chip design flow and highlighted the distinctions between macro placement and well-structured domains such as Go. These distinctions help illustrate why domain expertise remains crucial in macro placement tasks and why integrating expert knowledge is a necessary step toward building effective AI solutions in this domain.

4. **Generalization Experiments (Reviewer yPJt)**:
We performed more extensive generalization experiments, which confirm that EXPlace consistently outperforms existing RL-based methods across a variety of chip designs.

5. **Post-Routing Results and Ablation Studies (Reviewers yPJt and ocTp)**: We added post-routing results of power, area (for Reviewer yPJt) and routing violations (for Reviewer ocTp), and conducted more comprehensive ablation studies on peripheral biasing (for yPJt) and timing preference fine-tuning (for ocTp). These additions further strengthen the experimental validation of our method.

6. **Cost Function Design (Reviewer zEBD)**:
We discussed the cost-summing approach and provided the convergence curve of training costs, which supports the effectiveness of our cost function design.

7. **Examples of Layout Comparison (Reviewer zEBD)**: We provided layout visualizations as concrete case studies to illustrate the impact of each component of expert knowledge. These examples help clarify how expert guidance contributes to improved placement results.

We hope that the revisions and additional results have addressed your concerns and clarified the contributions of our work. Once again, we deeply appreciate your insightful feedback and look forward to any further comments you may have.

---

### Author Response · Authors · 2025-12-03
**Summary of Rebuttal (0/3)**

Dear AC,

Thank you very much for taking the time to review our paper and responses. To facilitate your decision-making, we have summarized the key concerns raised by each reviewer and our corresponding responses below:

## Summary of Reviewer yPJt
Reviewer yPJt (score 4) acknowledged that our core idea of integrating expert knowledge can enhance the RL placement method. The concerns are primarily about the comprehensiveness of our experiments, including additional runtime analysis (Weakness 1), sign-off power and area results (Weakness 2), additional ablation study (Weakness 3), and generalization test on additional cases (Weakness 4). In response, we have supplemented all the requested experiments to improve comprehensiveness, as detailed in the revised Appendix D.1 (ablation), D.2 (runtime), D.3 (generalization and runtime), and Table 2 (power and area).

### 1. Comparison results with runtime (Weakness 1)
The reviewer commented that our original runtime analysis is narrow and suggested benchmarking against advanced analytical and black-box optimization (BBO) placers. In response, we expanded the runtime analysis in Appendix D.2 and acknowledged that the runtime of case-by-case RL training is indeed longer than that of analytical and BBO methods. However, we would like to clarify that RL-trained models possess generalization capabilities to solve unseen cases without additional fine-tuning. **Under this generalization setting, the runtime efficiency of RL methods (including our EXPlace) is orders of magnitude higher than that of analytical and BBO placers, as shown in Table R1 below.** For example, EXPlace consumes 9.31s on superblue3, while LaMPlace [Geng et al., ICLR'25] costs 1368.00s.

Notably, the results also show that **the zero-shot EXPlace outperforms both BBO and analytical methods in terms of the key chip performance metrics, HPWL and WNS, while other RL-based placers fail to achieve such superiority in the zero-shot setting.**

Revisiting the case-by-case training setting, the longer runtime incurred by training does not imply that the significant performance gains of EXPlace shown in Tables 1 and 2 are less compelling. For both BBO and analytical methods, their results are obtained after full convergence, and providing additional time budgets may not yield further improvements. Moreover, the real-world design cycle of integrated circuits typically spans several weeks, which is more than sufficient to cover the cost of case-by-case RL training.





**Table R1**. Comparison under the generalization setting on the ICCAD 2015 benchmarks. Runtimes are reported in seconds. RL models trained on superblue1 are evaluated on other cases in a zero-shot manner. BBO and analytical methods are also included for comparison.

| Method         | Metric  | superblue1 | superblue3 | superblue4 | superblue5 | superblue7 | superblue10 | superblue16 | superblue18 | Avg. Rank |
|----------------|---------|------------|------------|------------|------------|------------|-------------|-------------|-------------|------------|
| DREAMPlace     | HPWL    | 768.5      | 971.3      | 673.7      | 824.3      | 1063.3     | 1200.9      | 665.1       | **256.8**   | 4.50       |
|                | WNS     | -67.50     | -106.86    | -67.35     | **-57.50** | -101.27    | -264.08     | -36.57      | -22.12      | 3.38       |
|                | TNS     | -1389.24   | -1187.80   | -1044.56   | **-673.01**| **-520.82**| **-2583.61**| **-1011.32**| **-52.48**  | **1.50**   |
|                | Runtime | 287.30     | 158.11     | 107.54     | 121.20     | 199.03     | 210.90      | 112.70      | 66.12       | 4.12       |
| LaMPlace       | HPWL    | 577.7      | 845.5      | 467.3      | **645.3**  | 924.9      | 1050.5      | 498.7       | 281.6       | 2.62       |
|                | WNS     | -56.37     | -174.94    | -44.75     | -158.91    | -88.09     | **-77.54**  | **-34.68**  | -47.41      | 2.88       |
|                | TNS     | -1536.77   | -1797.70   | -1053.89   | -1300.71   | -1602.31   | -3441.67    | -1527.76    | -311.15     | 3.38       |
|                | Runtime | 792.00     | 1368.00    | 396.00     | 2592.00    | 756.00     | 11376.00    | 36.00       | 180.00      | 4.88       |
| MaskPlace      | HPWL    | 733.8      | 822.8      | 505.7      | 784.5      | 981.4      | 1006.5      | 531.6       | 328.2       | 3.75       |
|                | WNS     | -77.13     | -98.61     | -48.88     | -192.60    | **-77.44** | -108.36     | -42.04      | -31.49      | 3.38       |
|                | TNS     | -1644.56   | -1475.26   | -1452.97   | -2003.23   | -1771.30   | -4191.77    | -1763.34    | -219.48     | 4.38       |
|                | Runtime | 1.64       | 2.70       | 3.88       | 2.28       | 5.93       | 3.46        | 2.59        | 3.11        | **1.00**   |

**(Due to space limitation, the final two rows will be presented in the subsequent block)**

---

> ### Author Response · Authors · 2025-12-03
> **Summary of Rebuttal (1/3)**
>
> **(The results presented below follow Table R1)**
> | Method         | Metric  | superblue1 | superblue3 | superblue4 | superblue5 | superblue7 | superblue10 | superblue16 | superblue18 | Avg. Rank |
> |----------------|---------|------------|------------|------------|------------|------------|-------------|-------------|-------------|------------|
> | MaskRegulate   | HPWL    | 600.9      | 761.2      | 432.4      | 710.1      | 910.0      | 1190.9      | 517.0       | 291.6       | 2.88       |
> |                | WNS     | -78.50     | **-86.66** | -53.46     | -148.92    | -93.97     | -77.59      | -39.06      | -31.34      | 3.12       |
> |                | TNS     | -1093.88   | -1330.60   | -1253.22   | -1414.67   | -1826.75   | -4904.24    | -1421.76    | -217.42     | 3.38       |
> |                | Runtime | 4.69       | 8.08       | 8.92       | 6.89       | 17.95      | 8.09        | 3.37        | 3.72        | 2.25       |
> | EXPlace (Ours) | HPWL    | **536.9**  | **698.9**  | **393.1**  | 666.2      | **871.8**  | **862.6**   | **472.8**   | **270.1**   | **1.25**   |
> |                | WNS     | **-51.25** | -101.04    | **-39.76** | -140.89    | -87.10     | -99.97      | -43.40      | **-21.77**  | **2.25**   |
> |                | TNS     | **-1070.72**| **-1063.76**| **-820.35**| -1462.69 | -1479.43   | -3444.93    | -1999.02    | -140.40     | 2.38       |
> |                | Runtime | 5.28       | 9.31       | 10.68      | 6.80       | 19.55      | 7.49        | 5.03        | 4.97        | 2.75       |
>
> ### 2. Generalization test on all 8 cases (Weakness 4)
>
> The reviewer noted the absence of generalization tests on three specific circuits. We believe this concern is also effectively addressed by the above Table R1, which provides generalization results for all 8 test cases, covering the three circuits in question. The detailed results and corresponding analysis have been included in the revised Appendix D.3.
>
> ### 3. Comparison of post-route power and area (Weakness 2)
> The reviewer suggested including post-route power and core area metrics for a comprehensive sign-off quality assessment. In response, we have updated Table 2 to include these power and area metrics, also reproduced as Table R2 below. The results demonstrate that EXPlace achieves the lowest average rank of 2.33 for both power consumption and cell area, outperforming all compared approaches. These improvements stem from EXPlace's ability to reduce buffer usage via I/O keepout constraints and dataflow-aware optimization, while its highly regular macro placement further contributes by enabling more balanced clock tree construction.
>
> Table R2: Partial comparison results on OpenROAD benchmarks. We report Power (W), Cell Area (×10⁴μm²), and average ranking.
>
> | **Method**         | **Metric**   | **ariane133** | **ariane136** | **bp**     | **bp_be**  | **bp_fe**  | **swerv_wrapper** | **Avg. Rank** |
> |--------------------|--------------|---------------|---------------|------------|------------|------------|-------------------|---------------|
> | **DREAMPlace**     | Power        | 0.4537        | 0.5182        | **0.4954** | 0.2025     | 0.2141     | 0.2701            | 4.00          |
> |                    | Cell Area    | **38.55**     | 40.04         | **52.72**  | 12.27      | 7.50       | 23.08             | 2.67          |
> | **Hier-RTLMP**     | Power        | 0.4481        | 0.4991        | 0.4985     | 0.2055     | 0.2130     | 0.2670            | 3.33          |
> |                    | Cell Area    | 38.88         | 40.06         | 53.35      | 12.34      | 7.43       | 23.31             | 4.00          |
> | **MaskPlace**      | Power        | 0.4512        | 0.4976        | 0.4957     | 0.2032     | 0.2152     | 0.2732            | 3.83          |
> |                    | Cell Area    | 38.58         | **39.81**     | 53.84      | 12.24      | 7.41       | 23.46             | 3.33          |
> | **MaskRegulate**   | Power        | 0.4494        | 0.5049        | 0.5501     | 0.2079     | 0.2120     | 0.2747            | 5.00          |
> |                    | Cell Area    | 39.69         | 40.80         | 53.80      | 12.52      | 7.29       | 23.25             | 4.67          |
> | **LaMPlace**       | Power        | 0.4477        | **0.4969**    | 0.5211     | 0.2033     | 0.2095     | **0.2656**         | 2.50          |
> |                    | Cell Area    | 38.86         | 40.26         | 54.00      | 12.39      | 7.37       | 22.91             | 4.00          |
> | **EXPlace (Ours)** | Power        | **0.4472**    | 0.4996        | 0.5038     | **0.2012** | **0.2083** | 0.2681            | **2.33**      |
> |                    | Cell Area    | 39.15         | 40.07         | 52.92      | **12.21**  | **7.27**   | **22.85**         | **2.33**      |

---

> ### Author Response · Authors · 2025-12-03
> **Summary of Rebuttal (2/3)**
>
> ### 4. Ablation study of peripheral biasing (Weakness 3)
> The reviewer pointed out the omission of an ablation study on peripheral biasing. In response, we have supplemented our results by including a configuration without peripheral biasing in the updated Table 5 of Appendix D.1. The results demonstrate that excluding peripheral biasing leads to a degradation in both HPWL and timing performance. This decline can be attributed to increased congestion in the central region of the chip when the biasing is removed.
>
> ## Summary of Reviewer ocTp
> Reviewer ocTp (score 6) acknowledged that the proposed approach is innovative and practical, and expressed satisfaction with the extensive experiments and the promising performance gains. However, reviewer ocTp still has some moderate concerns regarding technical contributions and additional runtime consumption.
>
> ### 1. Main contribution of this paper (Weakness 1)
>
> The reviewer commented that the main contribution of this paper is to add three more expert masks. In response, we would like to clarify that the primary contribution of this paper is the conceptual advancement of integrating human expert knowledge into RL frameworks to achieve industrial-level results, rather than merely the technical addition of masks. Since the publication of AlphaChip [1] in *Nature*, many follow-up works [2,3,4] have sought to improve RL-based placement methods through better representation learning or the design of dense reward signals. However, most of these approaches still operate under oversimplified objective formulations, such as minimizing HPWL, and allow the RL agent to generate placement solutions without considering high-level constraints. **Under such a misaligned formulation, no matter how powerful the AI (RL) algorithm is, it is doomed to fail in achieving industrial-grade results**. Therefore, at this stage of RL-based placement, we believe the incorporation of expert knowledge to enhance both the framework and its objective formulation **represents a fundamental and necessary step forward**. Our work takes an early step in this direction and demonstrates substantial empirical gains. We believe this conceptual contribution is both timely and impactful.
>
> [1] A graph placement methodology for fast chip design. Nature, 2021.
>
> [2] On joint learning for solving placement and routing in chip design. NeurIPS, 2021.
>
> [3] MaskPlace: Fast chip placement via reinforced visual representation learning. NeurIPS, 2022.
>
> [4] Reinforcement learning policy as macro regulator rather than macro placer. NeurIPS, 2024.
>
> ### 2. The computation of masks is very time-consuming (Weakness 2)
> Regarding the concern about runtime overhead, we believe the computational cost of mask generation in EXPlace is justified by the performance gains it yields. As shown in Table 7 of Appendix D.3 (reproduced as **Table R1** above), **EXPlace outperforms modern BBO and analytical placers with much less runtime. In contrast, while other RL methods with simpler formulations have faster inference, they fail to achieve such superior placement quality.**
>
> ### 3. Questions
> The reviewer ocTp also raised questions regarding implementation details, the ablation study on preference optimization, and the post-routing DRC results on the ICCAD 2015 benchmarks. In response, we have revised the paper to include a detailed description of the preference data, supplemented the results by directly comparing EXPlace with and without preference optimization (see Table 9), and conducted new experiments for post-routing PPA evaluation (see Tables 10 and 11). We believe these modifications effectively addressed the reviewer's questions.

---

> ### Author Response · Authors · 2025-12-03
> **Summary of Rebuttal (3/3)**
>
> ## Summary of Reviewer zEBD
> Reviewer zEBD (score 4) acknowledged the effectiveness and rationale of our proposed method. The main concern is that human knowledge may be a local optimum, and can only be beneficial for the early stage of learning, with AlphaGo and AlphaZero cited as supporting examples.
>
> ### 1. Response to weakness
> We believe the concern stated under 'Weakness' arises from a clear misunderstanding about the complexity inherent in chip placement. **Unlike well-structured games such as Go, chip placement is characterized by a substantially larger search space, black-box and computationally intensive evaluations, and intricate dependencies on upstream and downstream design stages.** Specifically, placement is only one local stage in the full physical design flow of integrated circuits, which also includes logic synthesis, floorplanning, timing optimization, clock tree synthesis, routing, and sign-off. These stages are closely nested, and optimizing one stage in isolation usually does not lead to satisfactory results in terms of power, performance, and area (PPA). The success of AlphaZero [1] benefits from efficient self-play and a reliable value network that guides exploration of the search space. However, when solving macro placement, it is extremely difficult to construct a value network that accurately models the impact of each action on the final PPA. Under such a “blind” setting, even a system as powerful as AlphaZero may struggle to perform effectively. Therefore, **integrating expert knowledge into AI-based placers is not just beneficial for the early stage of learning but essential for industrial deployment**. The expert priors we consider, grounded in long-standing industry experience, are aware of the trade-offs involved throughout the full design flow. They provide critical inductive biases and domain knowledge that can guide the learning process and reduce misalignment. In our work, we bridge this gap for the first time by integrating expert knowledge from EDA methods into the RL framework. We believe this is a practical and necessary advancement toward making post-AlphaChip [2] RL placement methods more relevant and effective for real-world chip design.
>
> [1] Mastering Chess and Shogi by Self-Play with a General Reinforcement Learning Algorithm. ArXiv, 2017.
>
> [2] A graph placement methodology for fast chip design. Nature, 2021.
>
> ### 2. Cost function (Questions 1 and 2)
> The reviewer also has concerns about the cost function design and how the cost empirically changes. In response, we have explained that **the linear-summing approach is widely used in existing EDA methods and related works provide guidance for configuring the trade-off weights**. The new results in Figure 10 of Appendix D.7 show that most training costs converge well, except that the I/O keepout cost fluctuates due to its assigned small weight. However, the layout visualization at the bottom of Figure 10 reveals that the policy, in fact, learns the knowledge of I/O keepout despite its unstable cost.
>
> ### 3. Layout comparison examples (Question 3)
> Regarding the reviewer's question of layout comparison examples, we direct the reviewer to the original Figures 8 and 9 that provide visual comparisons of EXPlace and other baselines. **These comparisons demonstrate that the intended effects of peripheral biasing, macro grouping, and I/O keepout constraints are successfully realized.**
>
> ------
> In summary, two of the three reviewers recognized the significant value of integrating expert knowledge into RL-based placement methods. While the third reviewer expressed reservations, we believe these stemmed from a misunderstanding of the chip placement context. We have provided comprehensive clarifications regarding the problem's complexity to resolve this. Other concerns raised were all moderate, mainly regarding the need for additional experiments. In response, we have supplemented all the requested experiments and analyses. Unfortunately, we did not receive any feedback from the reviewers by November 27th (when the system bug occurred). Nonetheless, we remain confident that our rebuttal and extensive new experiments have fully addressed all issues, and we believe reviewers will raise their scores after reviewing our response.
>
> We hope this summary is helpful for your evaluation. Please refer to our specific replies to reviewers and the revised paper for more details. Thank you very much again for your time.

---

### Meta-Review · Area_Chair_iki7 · 2026-01-06

**Summary:**

The paper proposes an RL-based macro placement framework that integrates expert knowledge and expert workflow imitation to better align RL optimization with industrial PPA metrics. Reviewers found the approach well-motivated and empirically strong. Main concerns were about runtime comparisons, missing sign-off metrics, incomplete ablations, and limited generalization evaluation, all of which were addressed in the rebuttal with substantial new experiments.

**Reviewer Concerns:**

yPJt: Runtime scope, post-route power/area, missing periphery ablation, and incomplete generalization were all fully addressed with new experiments.

ocTp: Concerns about novelty and mask overhead were largely addressed via clarified contributions, added ablations, and post-routing results.

zEBD: Empirical questions were addressed; remaining concern about expert knowledge limiting long-term innovation is philosophical rather than technical.

**Reviewer Scores:**

Reviewer yPJt: 4 → 5

Reviewer ocTp: 6 → 6

Reviewer zEBD: 4 → 4–5

---

### Decision · Program_Chairs · 2026-01-26

Accept (Poster)